# Cytoplasmic retention and degradation of a mitotic inducer enable plant infection by a pathogenic fungus

Paola Bardetti[1†], Sónia Marisa Castanheira[1‡], Oliver Valerius[2], Gerhard H Braus[2], José Pérez-Martín[1]*

[1]Instituto de Biología Funcional y Genómica (CSIC), Salamanca, Spain; [2]Department of Molecular Microbiology and Genetics, Institute for Microbiology and Genetics, Georg-August-University, Göttingen, Germany

**Abstract** In the fungus *Ustilago maydis*, sexual pheromones elicit mating resulting in an infective filament able to infect corn plants. Along this process a G2 cell cycle arrest is mandatory. Such as cell cycle arrest is initiated upon the pheromone recognition in each mating partner, and sustained once cell fusion occurred until the fungus enter the plant tissue. We describe that the initial cell cycle arrest resulted from inhibition of the nuclear transport of the mitotic inducer Cdc25 by targeting its importin, Kap123. Near cell fusion to take place, the increase on pheromone signaling promotes Cdc25 degradation, which seems to be important to ensure the maintenance of the G2 cell cycle arrest to lead the formation of the infective filament. This way, premating cell cycle arrest is linked to the subsequent steps required for establishment of the infection. Disabling this connection resulted in the inability of fungal cells to infect plants.
DOI: https://doi.org/10.7554/eLife.48943.001

*For correspondence:
jose.perez@csic.es

Present address: †Department of Biology, New York University, New York, United States; ‡Centro Nacional de Biotecnología (CSIC), Madrid, Spain

Competing interests: The authors declare that no competing interests exist.

## Introduction

Sexual reproduction is widely conserved within the eukaryotic life tree. The final outcome of this process is the fusion of two distinct haploid nuclei into a single diploid nucleus. To avoid an imbalance in the nuclear genetic information provided by each mating partner, the cell cycle status of both nuclei should be the same before karyogamy. In metazoans, this synchronization often occurs once the two partner nuclei are in the same zygotic cytoplasm (*Austin, 1978*). However, in simple eukaryotes such as fungi or unicellular algae, cell cycle synchronization occurs before cell fusion (*Hartwell, 1973*). In these organisms, cell cycle synchronization is mediated by the recognition of signals, often pheromones secreted by distinct mating partners. The paradigmatic and best studied case is the budding yeast *Saccharomyces cerevisiae*. In this fungus, pheromone recognition is mediated by plasma membrane-located receptors, which transmit the signal toward the cell cycle machinery using a widely conserved MAP kinase cascade (*Bardwell, 2005*). Pheromone recognition in budding yeast results in G1 cell cycle arrest, which is maintained throughout all cell fusion process, resulting in a diploid zygote that is able to resume the cell cycle or enter into meiosis (*Elion, 2000*). Premating cell cycle synchronization at G1 phase seems to be the rule, as other fungi such as *Schizosaccharomyces pombe* (*Davey, 1998*), diatoms (*Moeys et al., 2016*), and -most likely- algae such as *Chlamydomonas reinhardtii* (*Joo et al., 2017*) and the slime mold *Dictyostelium discoideum* (*Ishida et al., 2005*) apply the same principle. However, there is one exception in the fungal maize smut pathogen *Ustilago maydis*, where premating cell cycle synchronization in response to secreted sexual pheromones occurs at G2 phase (*García-Muse et al., 2003*). Paradoxically, in spite of the distinct cell cycle stage for arrest, the elements involved in the transmission process (pheromone and receptors, MAPK cascade and transcription factors) are similar to those described in fungi that undergo arrest at G1

**eLife digest** Many fungi that cause diseases in plants need specialized structures to penetrate the plant's tissues. To form these structures, the fungus must carefully control when and where its cells divide. As in other organisms, the sequence of events that lead to a fungal cell dividing in two are known as the cell cycle. Progress through the distinct steps in the cell cycle is regulated by enzymes including many that add or remove phosphate groups on other proteins. It remains unclear which regulatory enzymes allow any plant-infecting fungus to control its cell cycle when it forms an infection structure, but one fungus that could help answer this question is *Ustilago maydis*, the cause of a disease known as corn smut.

The corn smut fungus forms infection structures after two different mating strains meet on the surface of the plant, stop dividing and then fuse. This implies that the cell cycles of both strains need to be coordinated to allow the fungus to infect the plant. The two strains recognize each other via chemical signals known as pheromones, and Bardetti et al. now show that pheromone recognition in the corn smut fungus results in an enzyme called Cdc25 being disabled, which in turn causes cell division to stop. Cdc25 is a phosphatase, meaning it removes phosphate groups from cell cycle regulators that are found in the nucleus of the cell. Specifically, Cdc25 targets phosphate groups that would otherwise inhibit the activity of these proteins. Further experiments showed that, following pheromone recognition, Cdc25 is disabled via a two-step process: first it is prevented from entering the nucleus which keeps it away from its targets, and then it is degraded. Bardetti et al. went on to show that this last step was required for the fungus to infect corn plants, since interfering with the breakdown of Cdc25 impairs its ability to stop the cell cycle and form an infection structure.

Entry into plant tissue is a critical step for any parasites looking to invade a plant. Since it is difficult to reach the interior of plants with pesticides, most antimicrobial treatments in plants aim at prevention rather than cure. This means that increasing the delay between a fungus recognizing the surface of a plant and penetrating its tissues could give more time to prevent infections. These new findings represent a step towards achieving that goal, though more research is needed to better understand the molecular mechanisms required for the formation of infection structures in plant-infecting fungi.

DOI: https://doi.org/10.7554/eLife.48943.002

phase (*Müller et al., 2003*; *Vollmeister et al., 2012*). This result strongly suggested that the differences were linked to alternative wiring of the *U. maydis* pheromone MAPK cascade with cell cycle regulators, although these connections were largely unknown.

The reasons for the distinct cell cycle response to pheromone in *U. maydis* are likely related to the unusual developmental steps that mating triggers in this fungal system. In *U. maydis*, virulence and sexual development are intricately interconnected because the mating of two compatible budding haploid cells is the prerequisite to induce the infectious stage (*Brefort et al., 2009*; *Vollmeister et al., 2012*). Pathogenic development is mediated by two independent loci: the *a*-locus, which encodes a pheromone-receptor system, and the *b*-locus, which encodes a pair of homeoproteins (bW and bE). On the plant surface, infection is initiated upon the recognition of mating pheromone secreted by haploid cells of the opposite mating type (*Bölker et al., 1992*). This recognition induces G2 cell cycle arrest as well as the formation of long conjugation tubes (*García-Muse et al., 2003*; *Spellig et al., 1994*), which grow toward each other and fuse at their tips (*Snetselaar et al., 1996*). Cytoplasmic fusion is not followed by karyogamy, resulting in a dikaryotic cell. After cell fusion on the plant surface, the G2 cell cycle arrest is sustained, and the single dikaryotic cell grows in a polar manner, producing the infective filament. This hypha expands, accumulating the cytoplasm at the tip of the filament, whereas the distal parts of the hypha become vacuolated and are sealed off by the insertion of regularly spaced septa, resulting in the formation of characteristic empty sections (*Steinberg et al., 1998*). This growth mode enables the fungus to progress along the plant surface, most likely to find an appropriate point of entry. Eventually, the hyphae stop polar growth in response to an as yet unidentified signal, and their tips swell to form appressoria and penetrate the cuticle (*Snetselaar and Mims, 1992*; *Snetselaar and Mims, 1993*).

Once the filament enters the plant, the cell cycle is reactivated, and the fungus proliferates inside the plant.

During the differentiation process resulting in plant penetration, the presence of a sustained G2 cell cycle arrest is mandatory and the impairment of this cell cycle arrest resulted in the inability of the fungus to infect plants (*Castanheira and Pérez-Martín, 2015*). This cell cycle arrest is imposed first by the activation of the pheromone cascade and then maintained during the growth of the infective filament by a transcriptional regulator called b-factor, which is encoded in the *b*-locus and composed of two subunits (bW and bE), provided by each mating partner. The mechanisms involved in this sustained cell cycle arrest have been described only partially. While the manner by which the presence of b-factor arrests the cell cycle is comprehended in detail, the molecular intricacies associated with the cell cycle arrest induced in response to pheromone are still unknown.

G2/M transition in *U. maydis* is regulated by the presence of two distinct cyclin-dependent kinase (CDK) complexes: Cdk1-Clb1 and Cdk1-Clb2 (*Garcia-Muse, 2004*). Of these, the limiting step is provided by the activity of the Cdk1-Clb2 complex, which is controlled by the inhibitory phosphorylation of Cdk1. The level of this phosphorylation depends on the relative activity of the Wee1 kinase (which inhibits Cdk1) and the Cdc25 phosphatase (which activates Cdk1) (*Sgarlata and Pérez-Martín, 2005a*; *Sgarlata and Pérez-Martín, 2005b*). Not surprisingly, the mechanism by which the b-factor arrests the cell cycle at G2 during the growth of the dikaryotic infective filament relies on the increase of Cdk1 inhibitory phosphorylation: The b-factor activates the DNA damage response (*de Sena-Tomás et al., 2011*; *Mielnichuk et al., 2009*) in the absence of DNA damage (*Tenorio-Gómez et al., 2015*), resulting in the phosphorylation of Cdc25, promoting thereby its interaction with 14-3-3 proteins, which in turn inactivates the phosphatase by its retention in the cytoplasm (*Mielnichuk and Pérez-Martín, 2008*); at the same time, the b-factor represses the transcription of *hsl1*, which encodes a kinase that downregulates Wee1 kinase, increasing as a consequence the level of inhibitory phosphorylation of Cdk1 (*Castanheira et al., 2014*). Moreover, a second cell cycle brake is added during the formation of the infective filament since the b-factor also activates the transcription of *biz1*, a transcriptional regulator that represses the transcription of *clb1*, encoding the b cyclin required for the second Cdk1-cyclin complex involved in G2/M transition (*Flor-Parra et al., 2006*; *Garcia-Muse, 2004*).

Here we tried to uncover the elements required for the pheromone-induced cell cycle arrest. Although it was possible that the pheromone response shared the same regulatory scheme as the b-induced cell cycle arrest, we show that this is not entirely the case and that some of the elements involved are different. We report that the pheromone response MAPK cascade is distinctly wired to cell cycle regulation, resulting first in the inhibition of the nuclear localization of Cdc25, via importin phosphorylation, and second, once the MAPK signaling reaches a threshold, in the degradation of the accumulated Cdc25. These steps were found to be required to ensure the maintenance of cell cycle arrest during the infection process. Inability to do so resulted in a strong defect in virulence.

## Results

### Activation of the pheromone cascade promotes G2 cell cycle arrest via inhibitory phosphorylation of Cdk1

We sought to address the molecular mechanisms behind the G2 cell cycle arrest observed upon pheromone response in *U. maydis*. This response requires, in each mating partner, the recognition of the compatible pheromone by its cognate receptor and the transmission of the signal through a conserved MAPK cascade (*Müller et al., 2003*; *Vollmeister et al., 2012*). However, the expression of pheromone and pheromone receptor genes requires poor nutritional conditions, which enables the activity of the transcriptional regulator Prf1 that activates the promoters from the *a*-locus (*Hartmann et al., 1999*; *Kaffarnik et al., 2003*) (*Figure 1—figure supplement 1A*). Since changes in nutritional conditions could alter the cell cycle pattern in this fungus (*Pérez-Martín et al., 2006*), we took advantage of the previous description of an activated allele of the pheromone cascade MAPKK Fuz7 (*fuz7DD*, *Figure 1—figure supplement 1B*), whose expression faithfully recapitulates the pheromone response in *U. maydis* (*Müller et al., 2003*; *Zarnack et al., 2008*). In this way, we make the activation of the pheromone MAPK cascade independent of the elements located

upstream of this cascade (*i. e.* receptors and pheromones) allowing us to focus on the connections between the pheromone response MAPK cascade and cell cycle regulators.

When an ectopic copy of the *fuz7*[DD] allele was expressed under the control of the *crg1* promoter (induced by arabinose and repressed by glucose) (*Figure 1—figure supplement 1C and D*), it mimicked the G2 cell cycle arrest observed when pheromone is sensed by *U. maydis* (*García-Muse et al., 2003*): cells accumulate 2C DNA content, carrying a single nucleus with an intact nuclear membrane (*U. maydis* breaks down its nuclear envelope at mitosis; *Straube et al., 2005*) (*Figure 1A and B*). Furthermore, this cell cycle arrest was dependent on Kpp2, the downstream MAPK, but independent of Prf1 (*Figure 1—figure supplement 1E*).

We observed that the expression of the *fuz7*[DD] allele was correlated with an increase in the level of inhibitory phosphorylation of Cdk1, which has been reported to be associated with G2 cell cycle arrest in *U. maydis* (*Sgarlata and Pérez-Martín, 2005a*; *Sgarlata and Pérez-Martín, 2005b*) (*Figure 1C* and *Figure 1—figure supplement 2A*). Moreover, the impairment of Cdk1 inhibitory phosphorylation, either by the expression of *cdk1*[AF], an allele refractory to inhibitory phosphorylation, or by the downregulated expression of *wee1*, the cognate kinase responsible for inhibitory phosphorylation (*Sgarlata and Pérez-Martín, 2005b*), abrogated the *fuz7*[DD]-dependent cell cycle arrest (*Figure 1D* and *Figure 1—figure supplement 2B–F*).

These results supported the notion that the G2 cell cycle arrest associated with the activation of the pheromone cascade was dependent on the inhibitory phosphorylation of Cdk1.

## The molecular mechanisms for pheromone cascade-mediated cell cycle arrest are likely to be different from those described for b-dependent cell cycle arrest

The mechanism of cell cycle arrest induced by the b-factor, which is responsible for the sustained G2 arrest during the growth of the infective filament, also involves an increase in the level of Cdk1 inhibitory phosphorylation (*Mielnichuk et al., 2009*). Moreover, in agreement with a previous report (*Zarnack et al., 2008*), we observed that the transcription of *hsl1*, a negative regulator of the Wee1 kinase, was strongly downregulated upon expression of the *fuz7*[DD] allele (*Figure 1—figure supplement 3A*), as it occurs in b-dependent cell cycle arrest (*Castanheira et al., 2014*; *Heimel et al., 2010*). These findings prompted us to think that pheromone-dependent and b-dependent cell cycle arrests might share the same molecular mechanisms. To add further support to this idea, we analyzed the involvement in the *fuz7*[DD]-dependent cell cycle arrest of other elements required for b-dependent cell cycle arrest, like the Chk1 kinase. However, our results indicated that Chk1 was not involved in *fuz7*[DD]-dependent cell cycle arrest (*Figure 1—figure supplement 3B*). In the same way, it was already reported that expression of the *fuz7*[DD] allele does not upregulate *biz1* expression (*Flor-Parra et al., 2006*; *Zarnack et al., 2008*), suggesting different mechanisms for pheromone- and b-dependent cell cycle arrest.

The target of Cdk1 inhibitory phosphorylation in *U. maydis* is the Cdk1-Clb2 complex, which is the main regulatory control for G2/M transition (*Garcia-Muse, 2004*; *Sgarlata and Pérez-Martín, 2005b*). The amount of Cdk1 inhibitory phosphorylation depends on the relative activity levels of the Wee1 kinase and the Cdc25 phosphatase (*Sgarlata and Pérez-Martín, 2005a*). We analyzed the effects of expression of the *fuz7*[DD] allele on the levels of these regulators (*Figure 1E*). We observed that the protein levels of Cdc25 dropped abruptly upon expression of the *fuz7*[DD] allele, although this decrease in Cdc25 levels cannot be attributed to a decrease in the mRNA levels of its gene (*Figure 1—figure supplement 3A*).

Since the observed downregulation of Cdc25 levels could account for G2 cell cycle arrest (*Sgarlata and Pérez-Martín, 2005a*), we aimed to bypass the *fuz7*[DD]-dependent cell cycle arrest by overexpression of *cdc25* at the same time as expression of the *fuz7*[DD] allele. However, contrary to our expectations, overexpression of an ectopic copy of *cdc25* does not abrogate cell cycle arrest, in spite of the presence of high protein levels of Cdc25 in these conditions (*Figure 1F* and *Figure 1—figure supplement 4*). This result not only strongly suggested that the decrease of Cdc25 levels was not the cause of the cell cycle arrest, but it also supported our view of distinct molecular mechanisms between pheromone- and b-factor-induced cell cycle arrest: While high levels of Cdc25 do not affect the ability to arrest the cell cycle by the pheromone cascade activation, in the b-dependent cell cycle arrest, Cdc25 is retained at the cytoplasm by Bmh1 (14-3-3 protein), and this repression can be overwhelmed by high levels of Cdc25 (i. e. overexpressing Cdc25) (*Mielnichuk et al., 2009*).

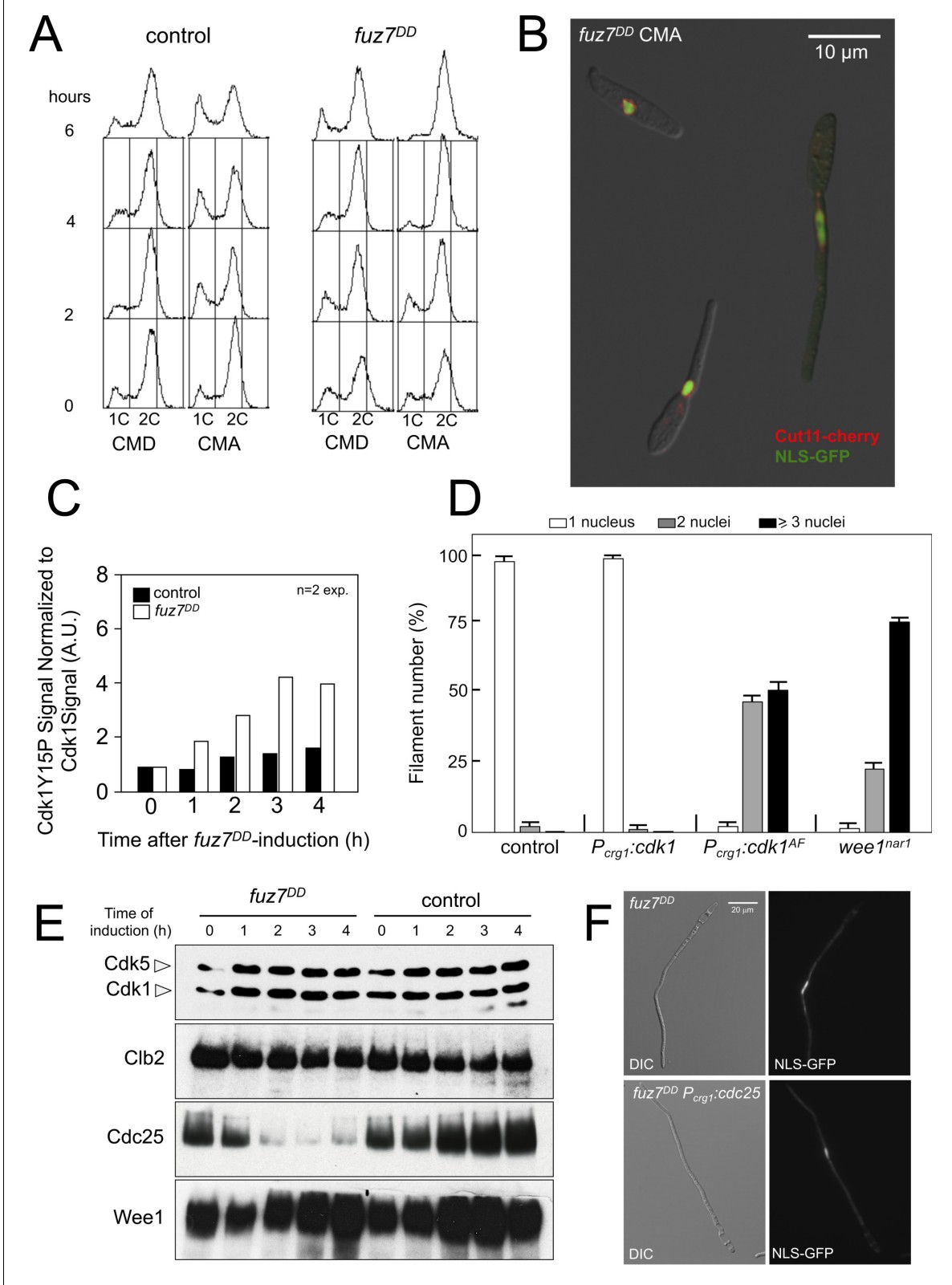

**Figure 1.** Expression of *fuz7$^{DD}$* allele promotes a G2 cell cycle arrest that depends on Cdk1 inhibitory phosphorylation. (**A**) Cells expressing the *fuz7$^{DD}$* allele accumulated with a 2C DNA content. Fluorescence/Activated Cell Sorter (FACS) analysis of the DNA content of a control strain and a strain carrying an ectopic copy of the *fuz7$^{DD}$* allele under the control of the *crg1* promoter growing in inducing (Complete Medium Arabinose, CMA) and non-inducing (Complete Medium Glucose, CMD) conditions (**Figure 1—figure supplement 1**). The period of incubation in testing media is indicated

*Figure 1 continued on next page*

*Figure 1 continued*

(hours). **(B)** Cells expressing the *fuz7DD* allele induce conjugative hyphae that are arrested in G2 phase. Representative image of cells expressing the *fuz7DD* allele and carrying NLS-GFP and Cut11-Cherry fusions to detect the nucleus and the nuclear envelope, growing in CMA for 6 hr. This image was a composition from various images to show different stages during the production of the conjugation hyphae. Bar: 15 µm. **(C)** Cells expressing the *fuz7DD* showed increased levels of Cdk1 inhibitory phosphorylation (Cdk1Y15P). Data acquisition is described in *Figure 1—figure supplement 2A* and. Means are shown (*Figure 1—source data 1*). **(D)** Interfering with the Cdk1 inhibitory phosphorylation resulted in inability to arrest cell cycle upon *fuz7DD* allele expression. Fuz7DD-derived strains carrying the *NLS-GFP* reporter as well as the indicated mutations were incubated in inducing conditions (CMA) for 6 hr. Filaments were sorted as carrying 1, 2 or 3 and more nuclei. The graph shows the result from three independent experiments, counting more than 100 filaments each. Means and SDs are shown (*Figure 1—figure supplement 2* and *Figure 1—source data 2*). **(E)** Protein levels of G2/M regulators upon *fuz7DD* allele expression. Strains carrying HA-tagged versions of Clb2, Cdc25 and Wee1 and carrying the *fuz7DD* allele or not (control) were incubated for the indicated time in induction conditions (CMA). Similar amount of protein extracts was separated by SDS-PAGE. Immunoblots were incubated with an antibody against HA. As loading control, we used the Cdk1 protein, which can be detected using anti-PSTAIRE (which recognizes both Cdk1 and Cdk5). **(F)** Overexpression of *cdc25* does not abrogate the *fuz7DD*-dependent cell cycle arrest. Representative images of cultures growing in inducing conditions (CMA) for 6 hr, from a control strain expressing the *fuz7DD* allele, and a strain co-expressing both the *fuz7DD* allele and an ectopic copy of *cdc25* (*Figure 1—figure supplement 4*).
DOI: https://doi.org/10.7554/eLife.48943.003

The following source data and figure supplements are available for figure 1:

**Source data 1.** Data for *Figure 1C*.
DOI: https://doi.org/10.7554/eLife.48943.008

**Source data 2.** Data for *Figure 1D*.
DOI: https://doi.org/10.7554/eLife.48943.009

**Figure supplement 1.** Activation of the pheromone response cascade upon expression of the *fuz7DD* allele.
DOI: https://doi.org/10.7554/eLife.48943.004

**Figure supplement 2.** Fuz7DD-dependent cell cycle arrest requires inhibitory phosphorylation of Cdk1.
DOI: https://doi.org/10.7554/eLife.48943.005

**Figure supplement 3.** The mechanism of *fuz7DD*-dependent cell cycle arrest is unrelated to b-dependent cell cycle arrest.
DOI: https://doi.org/10.7554/eLife.48943.006

**Figure supplement 4.** Overexpression of *cdc25* does not abrogate the *fuz7DD*-dependent cell cycle arrest.
DOI: https://doi.org/10.7554/eLife.48943.007

In summary, these previous results indicated that some elements, like the downregulation of *hsl1* (and thereby the upregulation of Wee1) seemed to be shared by b- and pheromone cascade-induced cell cycle arrest. However, other elements, like the downregulation of *biz1* and the Chk1-mediated retention of Cdc25 at cytoplasm seemed to be unique for b-dependent cell cycle arrest.

## Pheromone cascade-induced cell cycle arrest depends on an alternative cyclin interacting with a Cdk-like kinase

In our search for elements connected to the pheromone cascade that could be involved in the induction of the cell cycle arrest, we recalled the gene *pcl12*, which has been reported to be strongly induced by the pheromone MAPK (*Flor-Parra et al., 2007*). This gene encodes a cyclin from the Pcl family (*Measday et al., 1997*), and its ectopic expression under a regulatable promoter (such as the *crg1* promoter) was sufficient to induce the formation of a cell structure resembling a conjugation tube, which was also arrested at G2 phase (*Flor-Parra et al., 2007*). For that reason, we were curious about the involvement of this protein in the pheromone cascade-induced cell cycle arrest. Indeed, we found that Pcl12 was required for the induction of cell cycle arrest as well as for the observed decrease in Cdc25 levels upon the expression of the *fuz7DD* allele (*Figure 2A and B*).

Pcl12 forms a complex with the essential cyclin-dependent kinase Cdk5 (*Castillo-Lluva et al., 2007*). The Pcl12-Cdk5 complex is required for sustained polar growth during the formation of the conjugation tubes in response to pheromone treatment (*Flor-Parra et al., 2007*). However, we found that Cdk5 was not required for *fuz7DD*-induced cell cycle arrest (*Figure 2—figure supplement 1*). Since Pcl12 is a cyclin, this result suggested the existence of alternative partners (most likely kinases) for Pcl12 during pheromone cascade-induced cell cycle arrest. To identify these putative partners, we performed coimmunoprecipitation coupled with liquid chromatography-mass spectrometry (LC/MS) analysis of a GFP-tagged Pcl12 version in the presence of the expression of the *fuz7DD* allele. We found the kinase Crk1 among the major peptides copurifying with Pcl12 (*Figure 2—figure supplement 2*, *Figure 2—source data 1*). Crk1 (Cdk-Related Kinase 1) was previously described as

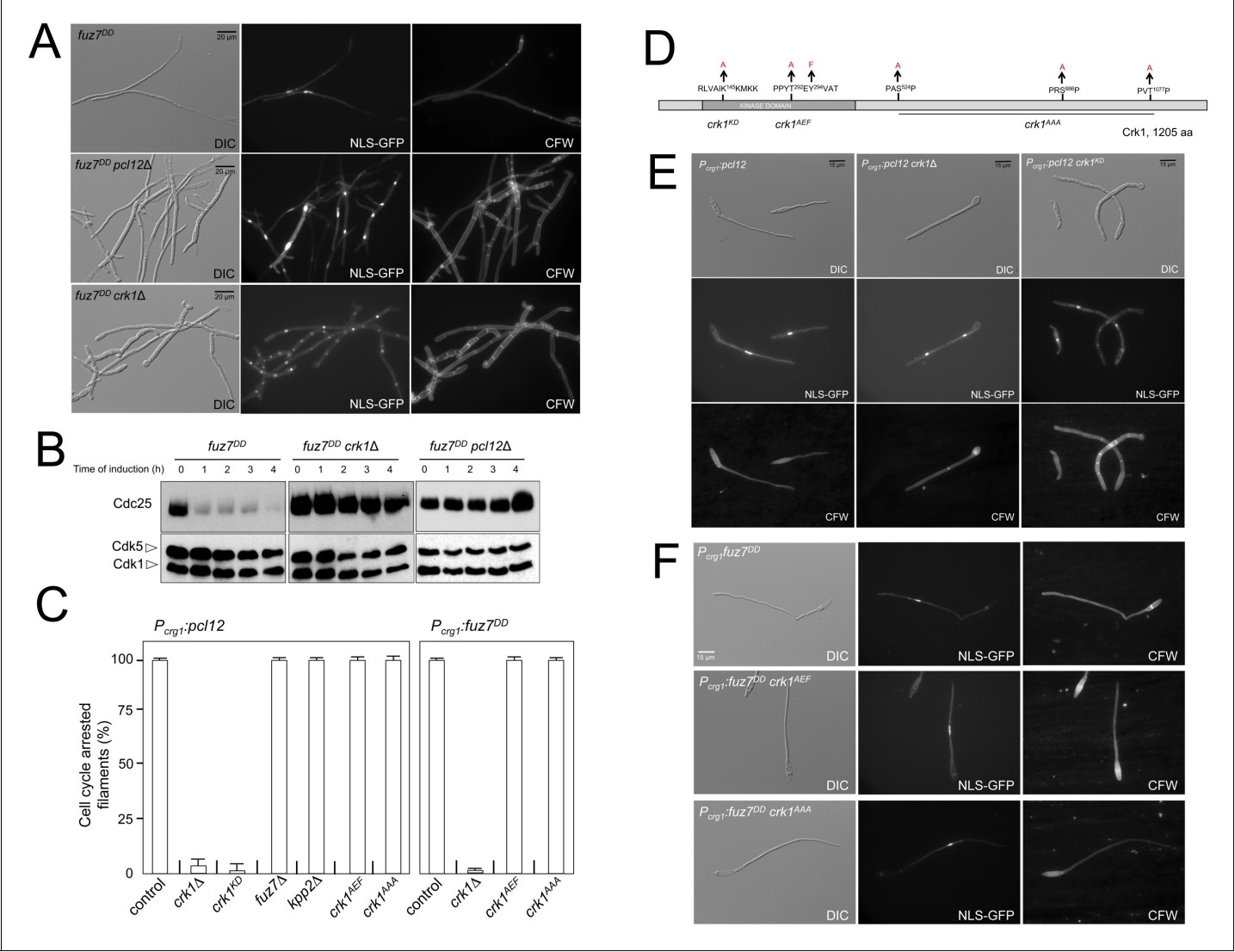

**Figure 2.** Fuz7$^{DD}$-induced cell cycle arrest depends on an alternative cyclin interacting with a Cdk-like kinase. (**A**) The cyclin Pcl12 and the Cdk-like kinase Crk1 were required for Fuz7$^{DD}$-dependent cell cycle arrest. Representative images of cultures of strains carrying the *fuz7$^{DD}$* allele and the indicated mutations. Cultures were incubated for 12 hr in inducing conditions for *fuz7$^{DD}$* (CMA). Cells carried a constitutively expressed NLS-GFP reporter to detect nuclei and were stained with Calcofluor White (CFW) to detect septa. Note that filaments in the mutants were composed of cell compartments carrying one nucleus each and separated by septa. Bar: 20 µm. (**B**) The cyclin Pcl12 and the Cdk-like kinase Crk1 were required for Fuz7$^{DD}$-dependent decrease of Cdc25 levels. Western blot analysis to show the level of Cdc25 (upper blot) upon expression of *fuz7$^{DD}$* allele in cells growing in inducing conditions (CMA) for the indicated time. Levels of Cdk1 were used as loading control (bottom blot). (**C**) Ability to arrest the cell cycle upon expression of *fuz7$^{DD}$* or *pcl12* in distinct mutant strains. The indicated strains, which also carried the *NLS-GFP* transgene, were incubated in inducing conditions (CMA) for 6 hr. Filaments from each culture were counted and sorted as carrying 1 (cell cycle arrested) or more than one nucleus (not arrested). The graph shows the result from three independent experiments, counting more than 100 filaments each. Means and SDs are shown (*Figure 2—source data 2*). Representative images corresponding to the respective cultures could be found at *Figure 2E and F* as well as at *Figure 2—figure supplement 3*. (**D**) Scheme of Crk1, showing the mutant alleles used in this work. These mutants were already described: *crk1$^{KD}$*, is a kinase-dead loss of function mutant; *crk1$^{AEF}$* is refractory to T-loop activation by Fuz7; *crk1$^{AAA}$* is refractory to phosphorylation by the MAPK Kpp2 (*Garrido et al., 2004*). (**E**) Crk1 is required for cell cycle arrest promoted upon expression of *pcl12*. Representative images of cultures of strains carrying an ectopic copy of *pcl12* under *crg1* promoter and the indicated mutations. Crk1$^{KD}$ carried the K145A mutation that inactivates its kinase catalytic activity (*Garrido et al., 2004*). Cultures were incubated for 6 hr in inducing conditions for *pcl12* (CMA). Cells carried a constitutively expressed NLS-GFP reporter to detect nuclei and were stained with Calcofluor White (CFW) to detect septa. Bar: 15 µm. (**F**) Representative images of cultures of strains carrying the *fuz7$^{DD}$* allele as well as the indicated mutations. Cultures were incubated for 6 hr in inducing conditions for *fuz7$^{DD}$* (CMA). Cells carried a constitutively expressed NLS-GFP reporter to detect nuclei and were stained with Calcofluor White (CFW). Bar: 15 µm.

DOI: https://doi.org/10.7554/eLife.48943.010

The following source data and figure supplements are available for figure 2:

*Figure 2 continued on next page*

*Figure 2 continued*

**Source data 1.** Data from LC/MS.
DOI: https://doi.org/10.7554/eLife.48943.015
**Source data 2.** Data for *Figure 2C*.
DOI: https://doi.org/10.7554/eLife.48943.016
**Figure supplement 1.** Cdk5 is not required for cell cycle arrest upon expression of the *fuz7$^{DD}$* allele.
DOI: https://doi.org/10.7554/eLife.48943.011
**Figure supplement 2.** Protein interacting with Pcl12-GFP.
DOI: https://doi.org/10.7554/eLife.48943.012
**Figure supplement 3.** Cell cycle induced by the ectopic expression of *pcl12* is independent on MAPK-mediated phosphorylation of Crk1.
DOI: https://doi.org/10.7554/eLife.48943.013
**Figure supplement 4.** Loss-of-function mutations in *crk1* and *pcl12* showed additive defects in morphology.
DOI: https://doi.org/10.7554/eLife.48943.014

a regulator of polar growth since its overexpression induces cell filamentation (*Garrido and Pérez-Martín, 2003*). In support of a functional role of the observed Pcl12 and Crk1 physical interaction, we found that loss of function of Crk1 abrogated *fuz7$^{DD}$*-dependent cell cycle arrest as well as the decrease in Cdc25 levels in a similar manner as the *pcl12* mutant did (*Figure 2A and B*). Furthermore, we also observed that Crk1 was also required for cell cycle arrest upon ectopic expression of *pcl12* (*Figure 2C and E*). These results sustained the idea that Crk1 and Pcl12 were working together during the pheromone cascade-induced cell cycle arrest, most likely as a complex.

Interestingly, Crk1 activation was previously described to be linked to the pheromone cascade: In order to promote hyperpolarized growth, Crk1 must be activated via phosphorylation of its T-loop by the MAPKK Fuz7, as well as by phosphorylation of the C-terminal end by the MAPK Kpp2 (*Garrido et al., 2004*). Therefore, we wondered whether these described connections between Crk1 and the pheromone cascade during the promotion of filamentous growth were also required for *fuz7$^{DD}$*-induced cell cycle arrest. To address this question, we took advantage of previously described mutant alleles of *crk1* refractory to phosphorylation by Fuz7 (*crk1$^{AEF}$*) or by Kpp2 (*crk1$^{AAA}$*) (*Figure 2D*) (*Garrido et al., 2004*). Much to our surprise, we found that cells carrying *crk1$^{AEF}$* or *crk1$^{AAA}$* mutant alleles were able to arrest the cell cycle upon the expression of the *fuz7$^{DD}$* allele (*Figure 2F*). Moreover, we also found that the ectopic expression of *pcl12* was still able to arrest the cell cycle in cells carrying different mutations affecting the phosphorylation of Crk1 by the MAPK cascade, such as the loss-of-function in *fuz7* or *kpp2*, or *crk1* alleles refractory to phosphorylation (*Figure 2C* and *Figure 2—figure supplement 3*).

These results indicated that the described activation of Crk1 via the pheromone cascade during filamentation played no role in the cell cycle arrest induced by the same MAPK cascade. To explain this apparent paradox, we propose that Crk1, which is a kinase that have features of both CDK-like and MAPK-like kinases (see discussion section for a detailed description of this hypothesis), can be activated by two distinct manners: either as a MAPK, by T-loop phosphorylation through the pheromone MAPK cascade, to support polar growth; or as a CDK, by interaction with Pcl12 cyclin, to support cell cycle arrest. Distinct activation mechanisms probably also involve distinct targets explaining the distinct outcomes.

Both Crk1 and Pcl12 were previously described as regulators involved in morphogenesis of the conjugative tube (*Flor-Parra et al., 2007*; *Garrido et al., 2004*), although the relationships between these factors at this level were not studied. We observed that, in contrast to the cell cycle arrest, which is abrogated by a single mutation in either *pcl12* or *crk1*, the effects of each gene mutation on the morphology of the resulting filament upon expression of *fuz7$^{DD}$* allele were distinct, and the double mutant was more affected than single mutants (*Figure 2—figure supplement 4A*). On the basis of these genetic interactions, we propose that Pcl12 and Crk1 work together, most likely forming a complex, and acting on the pheromone cascade-induced cell cycle arrest. However, it seems that these proteins also work in parallel pathways during the control of conjugation tube morphogenesis, most likely through the Pcl12-Cdk5 complex in one pathway, and through Crk1 receiving MAPK cascade signals in the other (*Figure 2—figure supplement 4B*).

## Kap123, the importin for Cdc25, seems to be phosphorylated upon activation of the pheromone cascade

The results shown above can be included in a working model in which the pheromone cascade-dependent induction of the expression of *pcl12* enables the formation of a Crk1-Pcl12 complex, which is responsible for the cell cycle arrest at G2. In support of this view, we observed that the cell cycle arrest induced by ectopic expression of *pcl12* was dependent on the inhibitory phosphorylation of Cdk1 (*Figure 3—figure supplement 1A and B*), similarly to the *fuz7^{DD}*-induced cell cycle arrest. However, in clear contrast with the observed drop in Cdc25 levels upon *fuz7^{DD}* expression, the levels of Cdc25 did not decrease upon the ectopic expression of *pcl12* (*Figure 3—figure supplement 1C*). This result reinforced our conclusion that the decrease in Cdc25 levels was not responsible for pheromone cascade-induced cell cycle arrest at G2.

Strikingly, we also observed that in the filaments produced upon ectopic *pcl12* expression, a GFP-Cdc25 fusion was excluded from the nucleus. Moreover, this exclusion was abrogated, as it was the cell cycle arrest, if the activity of Crk1 was eliminated (*Figure 3A and B*). Since the phosphatase Cdc25 must be transported to the nucleus to activate mitosis entry (*Mielnichuk and Pérez-Martín, 2008*), we hypothesized that retaining Cdc25 in the cytoplasm could explain the observed G2 cell cycle arrest upon ectopic *pcl12* expression and, most likely, also upon *fuz7^{DD}* expression. This hypothesis was reminiscent of the b-induced retention of Cdc25 at the cytoplasm via interaction with 14-3-3 proteins. However, we considered unlikely to be the same mechanism, because for b-dependent cell cycle arrest such cytoplasmic retention can be saturated by overexpression of Cdc25, and we observed that it was not the case for the pheromone cascade-induced cell cycle arrest (*Figure 1F*).

One of the interactors of Pcl12 obtained from the coimmunoprecipitation coupled with LC/MS analysis was the uncharacterized protein UMAG_15014. Sequence phylogeny analysis indicated that this protein (renamed Kap123) belongs to the family of β importins of class 3 and 4 (*Figure 3—figure supplement 2A*). Since Pcl12 is a cytoplasmic protein (*Flor-Parra et al., 2007*), we considered Kap123 unlikely to be involved in the transport of Pcl12 to the nucleus. A previous report showed that Sal3 from *S. pombe*, which is one of the members of the class 3 β importin family, was involved in the nuclear import of Cdc25 in this fungus (*Chua et al., 2002*). Therefore, we decided to analyze whether *U. maydis* Kap123 was required for the nuclear localization of Cdc25, and we found that it was (*Figure 3—figure supplement 2B and C*). This result prompted us to hypothesize that Kap123 could be a target of the Crk1-Pcl12 complex and that the action of this complex could disable the interaction between Kap123 and Cdc25 upon pheromone signaling. In this way, interference with the nuclear localization of Cdc25 could explain pheromone cascade-induced G2 cell cycle arrest. Using GFP-trap beads, we analyzed the ability of a Kap123-GFP fusion to interact with Cdc25 under conditions for the ectopic expression of *pcl12* in the presence or absence of functional Crk1 kinase. Additionally, to discard any effect of the expected interaction between Kap123 and Cdc25 as a consequence of the induced cell cycle arrest, we used the overexpression of Wee1 as a control for G2 cell cycle arrest (*Sgarlata and Pérez-Martín, 2005b*). In support of our hypothesis, we observed that the presence of a Crk1-Pcl12 complex, but not G2 cell cycle arrest alone, disables the ability of Kap123 to interact with Cdc25 (*Figure 3C*).

We also observed that upon *fuz7^{DD}* expression, the mobility of a Kap123-HA allele in SDS-acrylamide gels was reduced (*Figure 3D*), and this decrease in mobility can be eliminated by λ phosphatase treatment of the protein immunoprecipitates (*Figure 3E*). Moreover, we found similar electrophoretic mobility reduction upon ectopic expression of *pcl12* (*Figure 3F*).

The decrease in the electrophoretic mobility of Kap123 upon expression of *fuz7^{DD}* was dependent on the presence of functional alleles of *crk1*, *pcl12*, and *kpp2* (*Figure 3G*). In the case of ectopic expression of *pcl12*, Crk1 was required for the decreased electrophoretic mobility of Kap123, but neither *fuz7* nor *kpp2* were required (*Figure 3H*). These results mirrored the same genetic requirements as the cell cycle arrest, suggesting a causal relationship between phosphorylation of Kap123 and cell cycle arrest.

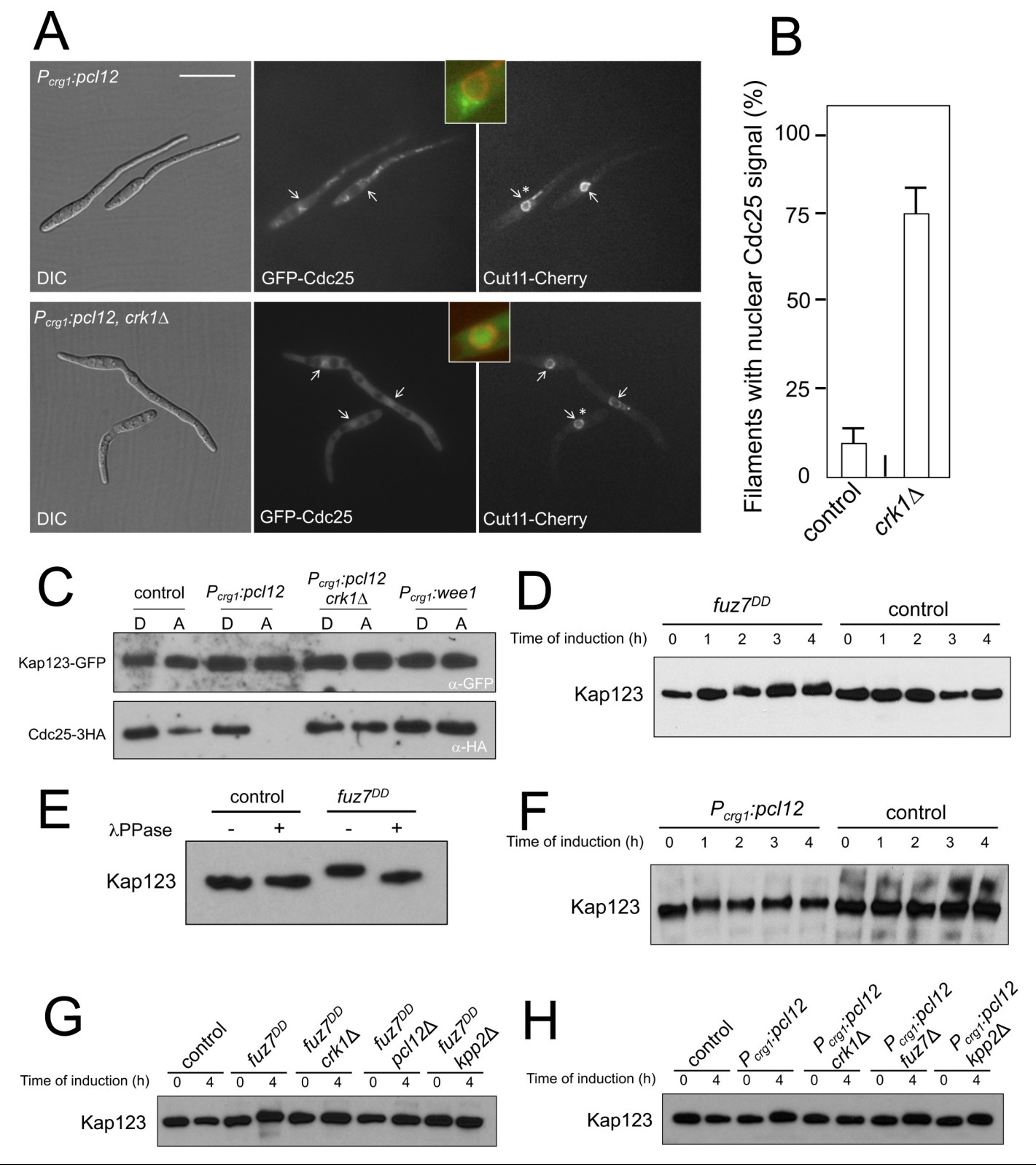

**Figure 3.** Kap123 seems to be phosphorylated upon activation of the pheromone cascade. (**A**) Expression of an ectopic copy of *pcl12* resulted in exclusion of Cdc25 from nucleus. Representative images of cultures of strains expressing an ectopic copy of *pcl12* and the indicated mutations. The cells carried an endogenous GFP-Cdc25 fusion as well as a Cut11-cherry fusion to detect nuclear membrane. Arrows pointed to nuclei in the filaments. Insets show merged images of selected nuclei (asterisk) from the respective filaments. Cultures were incubated for 6 hr in inducing conditions (CMA).
*Figure 3 continued on next page*

*Figure 3 continued*

Bar: 20 μm. (**B**) Quantification of number of filaments showing GFP fluorescence associated with the nucleus in control ($P_{crg1}$:*pcl12*) and *crk1* loss of function mutant (*crk1Δ*). The graph shows the result from three independent experiments, counting 50 filaments each (**Figure 3—source data 1**). Means and SDs are shown. (**C**) The presence of Crk1-Pcl12 complex inhibits the interaction between Cdc25 and its importin, Kap123. Soluble extracts from strains carrying Cdc25-3HA and Kap123-GFP tagged in their corresponding endogenous loci, and carrying ectopic copies of *pcl12* or *wee1* under the control of *crg1* promoter, were incubated with GFP-trap beads and the immunoprecipitates submitted to Western blot with anti-HA (Cdc25) and anti-GFP (Kap123) antibodies in succession. Cells were grown in inducing conditions (CMA, (**A**) or repressive conditions (CMD, (**D**) for *crg1* promoter during 6 hr. (**D and F**) Decrease in the electrophoretic mobility of Kap123 in response to pheromone-cascade activation. Extracts from cells carrying a Kap123-3HA allele and expressing *fuz7*$^{DD}$ or *pcl12* grown in inducing conditions (CMA) for the indicated time were submitted to Western blot with anti-HA. (**E**) The observed decrease in the electrophoretic mobility is sensitive to the treatment with phosphatase. Anti-HA immunoprecipitates of cell extracts from cultures expressing or not *fuz7*$^{DD}$ for 6 hr (CMA) were incubated at 30°C for 20 min in the absence (-) or presence (+) of lambda protein phosphatase (λ PPase) and were then subjected to immunoblot analysis with anti-HA. (**G**) Fuz7$^{DD}$-dependent phosphorylation of Kap123 requires Pcl12, Crk1 and Kpp2. Western blot analysis of extracts from strains carrying *kap123-3HA* allele and the indicated mutations, incubated in inducing conditions for *fuz7*$^{DD}$ expression (CMA) for the indicated time. (**H**) Ectopic expression of *pcl12* induces a decreased electrophoretic mobility of Kap123 that is dependent on Crk1 but not on Kpp2 and Fuz7. Western blot analysis of extracts from strains carrying *kap123-3HA* allele and the indicated mutations, incubated in inducing conditions for *pcl12* expression (CMA) for the indicated time.
DOI: https://doi.org/10.7554/eLife.48943.017

The following source data and figure supplements are available for figure 3:

**Source data 1.** Data for **Figure 3B**.
DOI: https://doi.org/10.7554/eLife.48943.020
**Figure supplement 1.** Cell cycle arrest upon ectopic expression of *pcl12* relies on Cdk1 inhibitory phosphorylation, although does not down-regulate Cdc25 levels.
DOI: https://doi.org/10.7554/eLife.48943.018
**Figure supplement 2.** Kap123 is the importin of Cdc25.
DOI: https://doi.org/10.7554/eLife.48943.019

## Pheromone cascade-induced cell cycle arrest is mediated by the phosphorylation of Cdc25 importin, Kap123

The previous results suggested that Kap123 could be phosphorylated by the Pcl12-Crk1 complex and that this modification could alter the ability of Kap123 to interact with Cdc25. To support this hypothesis, we looked for Kap123 mutants that were refractory to the phosphorylation by Pcl12-Crk1 complex. Since Crk1 has been defined both as a CDK-like and as a MAPK, and no consensus sequence for its phosphorylation sites was known, we considered, as a first approach, the established phosphorylation consensus sequences for both CDK and MAPK. Kap123 carries in its sequence 2 and 5 putative phosphorylation sites for CDK and MAPK, respectively. Using distinct threonine- or serine-to-alanine mutants at predicted phosphorylation sites in the Kap123 sequence (**Figure 4—figure supplement 1A**), we analyzed the electrophoretic mobility of HA-tagged Kap123 mutants in conditions of *fuz7*$^{DD}$ expression, and we found that the change of Thr867 to Ala (one of the putative CDK sites) resulted in abrogation of the electrophoretic mobility shift of Kap123 upon *fuz7*$^{DD}$ expression (**Figure 4—figure supplement 1B**). The mutant allele Kap123$^{T867A}$ showed no reduced electrophoretic mobility upon *fuz7*$^{DD}$ expression and *pcl12* expression (**Figure 4B and C**).

Encouragingly, the presence of this mutant allele makes the cells unable to undergo cell cycle arrest in response to the expression of either *fuz7*$^{DD}$ or *pcl12* (**Figure 4D and E**). Moreover, cells carrying the *kap123*$^{T867A}$ allele were unable to retain Cdc25 in the cytoplasm upon ectopic expression of *pcl12* (**Figure 4F and G**). Also, we found that the Kap123$^{T867A}$-GFP allele, in contrast to the wild-type GFP-tagged allele, was able to interact with Cdc25 in conditions of the ectopic expression of *pcl12* (**Figure 4H**).

Taken together, these data supported a model in which the pheromone cascade enables the formation of a Crk1-Pcl12 complex that inhibits the transport of Cdc25 to the nucleus by the phosphorylation of Kap123 importin, thereby promoting G2 cell cycle arrest.

## Cell cycle arrest in response to pheromone requires the phosphorylation of Kap123

Following our model, we aimed to test directly the response to pheromone of cells carrying mutations that abolished the pheromone cascade-induced cell cycle arrest. Addition of synthetic

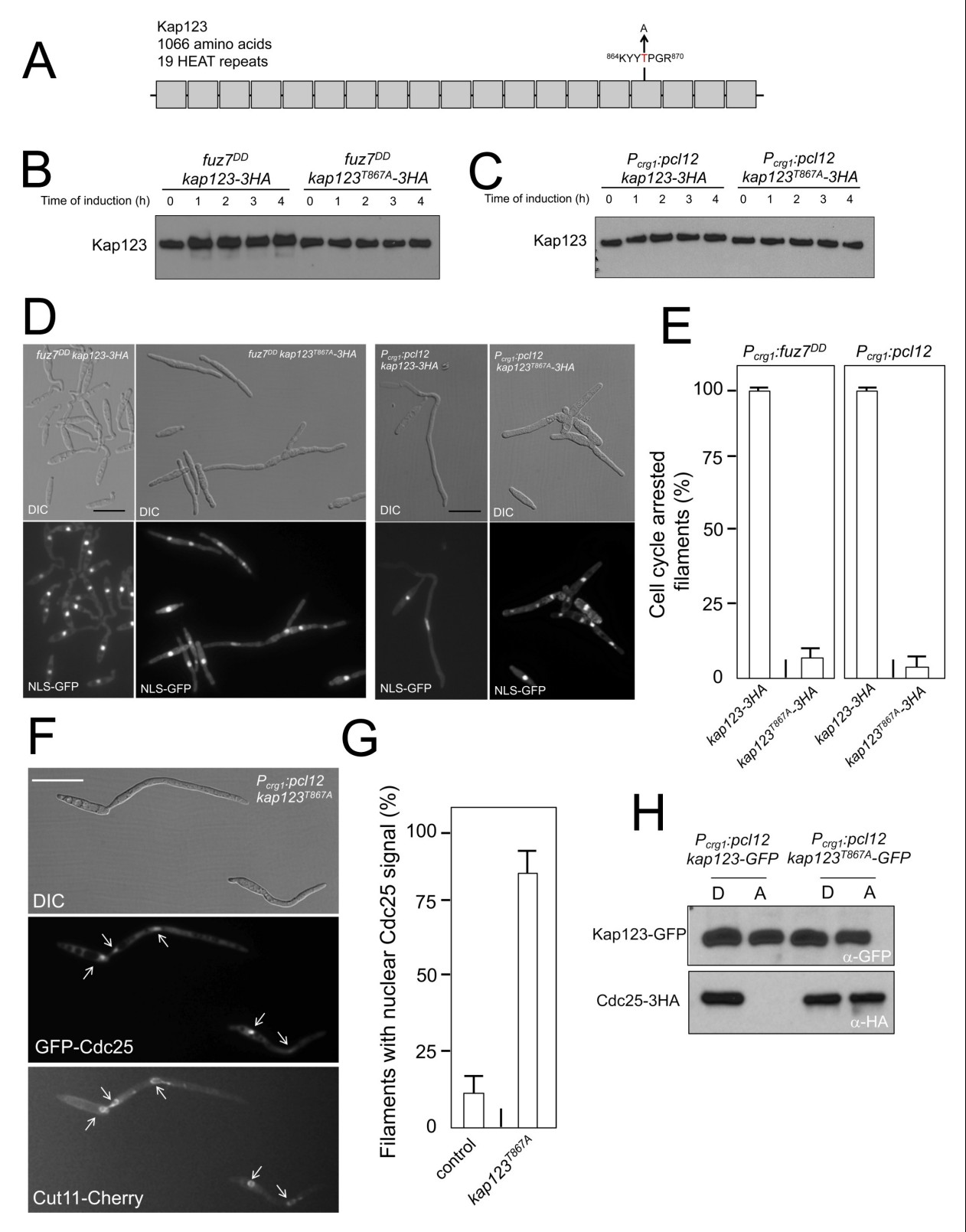

**Figure 4.** Pheromone cascade-induced cell cycle arrest is mediated by the phosphorylation of Cdc25 importin, Kap123. (**A**) Scheme of Kap123 showing the change to Alanine in Threonine 867, which is located at the 16th HEAT repeat. (**B and C**) Absence of electrophoretic mobility decrease in the Kap123T867A mutant upon expression of *fuz7DD* or *pcl12*. Extracts from cells carrying the indicated *kap123* allele and expressing *fuz7DD* or *pcl12* grown in inducing conditions (CMA) for the indicated time were submitted to Western blot with anti-HA. (**D and E**) Absence of cell cycle arrest in the

*Figure 4 continued on next page*

*Figure 4 continued*

Kap123[T867A] mutant. Control cells as well as cells carrying the mutant *kap123[T867A]* allele and expressing either *fuz7[DD]* or *pcl12* were incubated for 6 hr in CMA. Bar: 20 μm. Filaments from each culture were counted and sorted as carrying 1 (cell cycle arrested) or more than one nucleus (not arrested). The graph shows the result from three independent experiments, counting more than 100 filaments each (*Figure 4—source data 1*). Means and SDs are shown. (F) GFP-Cdc25 is not excluded from nucleus in the *kap123[T867A]* mutant strain. Representative image of strain expressing an ectopic copy of *pcl12* and carrying the *kap123[T867A]* allele. The cells also carried an endogenous GFP-Cdc25 fusion as well as a Cut11-cherry fusion to detect nuclear membrane. Arrows pointed to nuclei in the filaments. Cultures were incubated for 6 hr in inducing conditions (CMA). Bar: 20 μm. (G) Quantification of number of filaments showing GFP fluorescence associated with the nucleus in control and *kap123[T867A]* mutant cells expressing *pcl12* and incubated for 6 hr in inducing conditions (CMA). The graph shows the result from three independent experiments, counting 50 filaments each (*Figure 4—source data 2*). Means and SDs are shown. (H) Kap123[T867A] binds Cdc25 in the presence of a Crk1-Pcl12 complex. Soluble extracts from strains carrying Cdc25-3HA and Kap123-GFP or Kap123[T867A]-GFP alleles, and carrying ectopic copies of *pcl12* under the control of *crg1* promoter, were incubated with GFP-trap beads and the immunoprecipitates submitted to Western blot with anti-HA (Cdc25) and anti-GFP (Kap123) antibodies in succession. Cells were grown in inducing conditions (CMA, (A) or repressive conditions (CMD, (D) for *crg1* promoter during 6 hr.

DOI: https://doi.org/10.7554/eLife.48943.021

The following source data and figure supplement are available for figure 4:

**Source data 1.** Data for *Figure 4E*.
DOI: https://doi.org/10.7554/eLife.48943.023
**Source data 2.** Data for *Figure 4G*.
DOI: https://doi.org/10.7554/eLife.48943.024
**Figure supplement 1.** Analysis of the phosphorylation of Kap123.
DOI: https://doi.org/10.7554/eLife.48943.022

pheromone to a culture of compatible mating type cells induced the formation of conjugative tubes that were cell cycle arrested (*Spellig et al., 1994*; *Szabó et al., 2002*). Therefore, we treated *a1* mating type cells from *U. maydis* carrying an HA-tagged Kap123 allele, as well as its phosphorylation-defective version, with different concentrations of a2 synthetic pheromone, and analyzed the ability of cells to arrest the cell cycle. In concordance with our model, we found that cells carrying the *kap123[T867A]* allele were not able to arrest the cell cycle, even at the highest pheromone concentrations (*Figure 5A*). We analyzed, by Western blot, samples from these cultures and we observed a shift in the electrophoretic mobility of the wild-type allele but not the *kap123[T867A]* allele at different pheromone concentrations (*Figure 5B*). Furthermore, the presence of the *kap123[T867A]* allele makes the cells unable to keep Cdc25 out of the nucleus in response to synthetic pheromone (*Figure 5C and D*). All these results added further support to our model about the process by which pheromone triggers G2 cell cycle arrest in *U. maydis* cells.

## The absence of G2 cell cycle arrest during pheromone response has little impact on the ability of *U. maydis* to infect plants

The importance of cell cycle synchronization as a step prior to cell fusion during mating in unicellular eukaryotes was clearly illustrated more than 25 years ago by the defects in mating observed in the *S. cerevisiae far1* mutants, which lack the main control to arrest the cell cycle upon pheromone response (*Chang and Herskowitz, 1990*; *Pope et al., 2014*). Since the virulence of *U. maydis* is dependent on mating, we expected that the ability of *U. maydis* to infect corn plants would be affected in the absence of cell cycle arrest upon response to the pheromone. To test the hypothesis, we constructed compatible strains (*a1 b1* and *a2 b2* mating type) carrying the *kap123[T867A]* allele, and infected corn plants with compatible mixtures of control and mutant strains. We found that strains unable to undergo pre-mating cell cycle arrest were as virulent as wild-type strains (*Figure 6A*).

This unexpected result suggested a minor role of pheromone-induced cell cycle arrest in the virulence process, and therefore we were interested to understand the reasons for it. We wondered about the outcome from crosses of *kap123[T867A]* mutant strains. Mating in *U. maydis* is easily scored by the formation of dikaryotic filaments on charcoal plates, which can be observed as a white-appearing mycelial growth (fuzzy phenotype) (*Banuett and Herskowitz, 1989*). We observed that although slightly impaired, the mutant crosses showed a positive fuzzy phenotype (i.e., they were able to mate) (*Figure 6B*). Therefore, we were curious about the nuclear content of the filaments resulting from these crosses. To address this question, we crossed haploid strains expressing a GFP

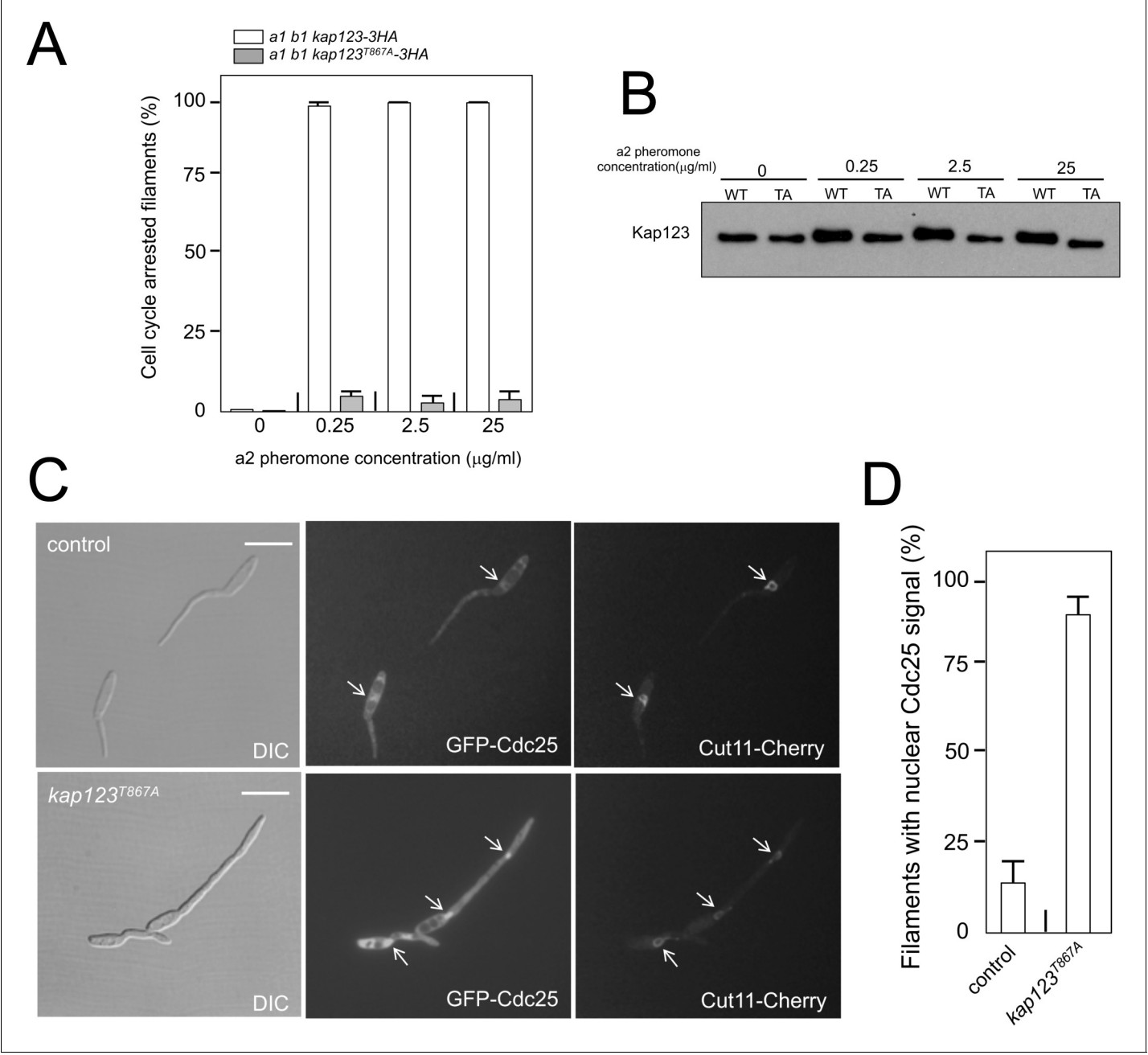

**Figure 5.** Cell cycle arrest in response to pheromone requires the phosphorylation of Kap123. (A) *kap123^T867A* mutant is unable to arrest the cell cycle in response to synthetic pheromone. The indicated strains were incubated for 6 hr in the presence of the indicated synthetic a2 pheromone concentrations in CMD medium. Filaments from each culture were counted and sorted as carrying 1 (cell cycle arrested) or more than one nucleus (not arrested). The graph shows the result from three independent experiments, counting more than 100 filaments each (*Figure 5—source data 1*). Means and SDs are shown. (B) Pheromone treatment results in a decrease in electrophoretic mobility of Kap123. Western blot from extracts obtained from *a1* mating-type cells carrying HA-tagged wild-type or *kap123^T867A* alleles that were incubated for 6 hr in the presence of the indicated synthetic a2 pheromone concentrations in CMD medium. (C) *kap123^T867A* mutant is unable to exclude Cdc25 from nucleus in response to synthetic pheromone. Representative images of cultures of wild-type and *kap123^T867A* mutant cells carrying endogenous GFP-Cdc25 and Cut11-Cherry gene fusions, in the presence of 0,25 μg/ml of a2 synthetic pheromone for 6 hr in CMD. Note the presence of more than one nucleus that accumulates GFP-Cdc25 in the *kap123^T867A* mutant strain. Bar: 15 μm. (D) Quantification of number of filaments showing GFP fluorescence associated with the nucleus in control and *kap123^T867A* mutant cells in the presence of 0,25 μg/ml of a2 synthetic pheromone for 6 hr in CMD. The graph shows the result from three independent experiments, counting 50 filaments each (*Figure 5—source data 2*). Means and SDs are shown.

DOI: https://doi.org/10.7554/eLife.48943.025

The following source data is available for figure 5:

*Figure 5 continued on next page*

*Figure 5 continued*

**Source data 1.** Data for *Figure 5A*.
DOI: https://doi.org/10.7554/eLife.48943.026
**Source data 2.** Data for *Figure 5D*.
DOI: https://doi.org/10.7554/eLife.48943.027

fusion to a nuclear localization signal under control of the *b*-dependent *dik6* promoter (*Mielnichuk et al., 2009*). In this way, only cells resulting from mating and therefore activating the transcriptional program dependent on b-factor (which subunits are provided by each mating partner) produced a fluorescent nuclear signal, so the mating-derived filaments can be distinguished from the cell population background (frequently enriched in aberrant elongated cells). We found that the majority of b-dependent filaments from wild-type crosses carried two nuclei, whereas filaments obtained from mutant crosses frequently carried two (62% of filaments), three (35%) and less frequently four (3%) nuclei (*Figure 6C and D*). We did not find filaments carrying more than four nuclei (from a total of 100 filaments counted).

To understand these results, it is worth remembering that once the cytoplasms of the compatible mating partners fuse, the formation of the heterodimeric b-factor activates the molecular mechanisms to induce a G2 cell cycle arrest. Indeed, we observed that the presence of the *kap123*$^{T867A}$ allele did not affect the ability of b-factor to arrest the cell cycle (*Figure 6—figure supplement 1*).

The inability of cells carrying the *kap123*$^{T867A}$ allele to arrest the cell cycle in response to pheromone (this is, to synchronize their cell cycles before cytogamy) will result, most likely, in the fusion of mating partners at different stages of the cell cycle. However, once the two cytoplasms fuse, the establishment of a barrier at G2 by the b-factor would resynchronize the respective nuclear content in the resulting filament. In the case of mating partners fusing their cytoplasms during G1 to G2 phases (i. e. providing a single nucleus each), the result will be a dikaryotic filament with nuclei arrested at G2. Only for those partners that fuse during or immediately after mitosis (and before cytokinesis), the result will be a filament carrying three nuclei (only one partner carried two nuclei before fusion) or four nuclei (the two partners carried two nuclei each), with these nuclei arrested at G2. In any case, the final result will be a cell cycle arrested filament (by virtue of b-factor) and therefore it will be proficient for infection, explaining the ability of mutant cells to infect corn plants. Interestingly, this ability to synchronize the cell cycle status of distinct nuclei in a common cytoplasm by homeodomain proteins of the b-factor family is the basis for accurate cell division in other basidiomycete dikaryotic fungi such as *Coprinopsis cinerea* (*de Sena-Tomás et al., 2013*).

## The MAPK Kpp2 directs Cdc25 downregulation

We were curious about the observed dramatic decrease in Cdc25 levels upon expression of the *fuz7*$^{DD}$ allele and whether this observation bears any functional relevance. To link the observed downregulation of Cdc25 with the pheromone response, we analyzed the levels of Cdc25 in *a1* mating type haploid cells in response to increasing concentrations of a2 synthetic pheromone (*Figure 7A*). We found that the lowest pheromone concentration tested (0.25 μg/ml), although able to induce the formation of cell cycle arrested conjugation tubes (*Figure 5A*), did not induce a decrease in the levels of Cdc25. However, increasing the amount of pheromone (up to 25 μg/ml) was correlated with a decrease in Cdc25 levels, suggesting a dose-response relationship between pheromone cascade signaling and Cdc25 levels.

We entertained the possibility that the observed downregulation of Cdc25 was a side effect of the permanent cytosolic retention of Cdc25, which might promote its degradation by some way, not necessarily related to the pheromone response. In support of this explanation we observed that the downregulation of Cdc25 levels requires the retention of Cdc25 in the cytoplasm, since expression of the *fuz7*$^{DD}$ allele in a *kap123*$^{T867A}$ mutant strain does not result in a decrease in Cdc25 levels (*Figure 7B*). However, we also observed that *pcl12* ectopic expression, despite being able to arrest the cell cycle and retain Cdc25 in the cytoplasm, was unable to induce a decrease in the Cdc25 level (*Figure 3—figure supplement 1C*). Furthermore, from the above described experiments using different pheromone doses, it was clear that the lowest pheromone concentration (0.25 μg/ml), was able to induce the phosphorylation of Kap123 (*Figure 5B*) as well as the retention of Cdc25 at

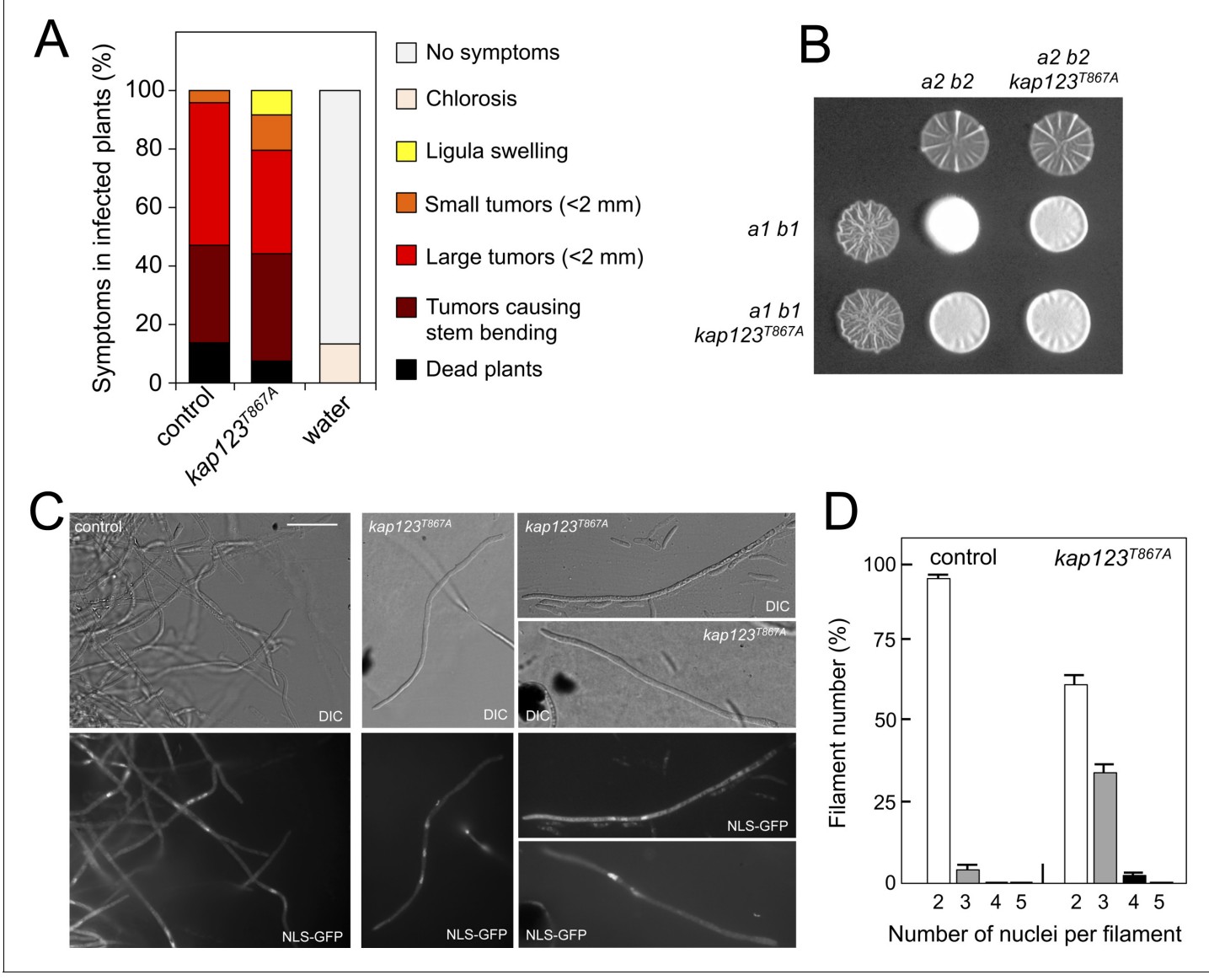

**Figure 6.** Effects of the absence of cell cycle arrest in the ability to mate and to infect plants. (A) *kap123^T867A* mutant strains are able to infect plants. Disease symptoms caused by crosses of wild-type and *kap123^T867A* mutant strains. The symptoms were scored 14 days after infection. Two independent experiments were carried out and the average values are expressed as percentage of the total number of infected plants (n: 30 plants in each experiment) (*Figure 6—source data 1*). (B) *kap123^T867A* mutant is able to mate. Crosses of control as well as *kap123^T867A* mutant strains carrying compatible mating types (*a1 b1* and *a2 b2*) in charcoal-containing agar plates. Positive fuzzy phenotype can be detected as a white-appearance mycelial growth. Note that mutant combinations were slightly affected in the ability to produce fuzzy phenotype. Plates were incubated at 22°C for two days. (C) *kap123^T867A* mutant filaments were affected in nuclear number. Crosses from compatible strains (wild-type or *kap123^T867A* mutant) carrying a NLS-GFP fusion under control of the b-factor-dependent *dik6* promoter were scrapped from agar surface, mounted on microscopy slides and epifluorescence was observed. Representative images show DIC and fluorescence in GFP channel. Bar: 20 µm. (D) Quantification of the nuclear content of filaments obtained from charcoal plates. The graph shows the result from two independent experiments, counting more than 50 filaments each (*Figure 6—source data 2*). Means and SDs are shown.

DOI: https://doi.org/10.7554/eLife.48943.028

The following source data and figure supplement are available for figure 6:

**Source data 1.** Data for *Figure 6A*.
DOI: https://doi.org/10.7554/eLife.48943.030
**Source data 2.** Data for *Figure 6D*.
DOI: https://doi.org/10.7554/eLife.48943.031
**Figure supplement 1.** The presence of the *kap123^T867A* allele did not affect the ability of b-factor to arrest the cell cycle.

*Figure 6 continued on next page*

*Figure 6 continued*

DOI: https://doi.org/10.7554/eLife.48943.029

cytoplasm (*Figure 5C and D*), but not its downregulation (*Figure 7A*). These observations suggested that although cytoplasmic retention was required, it was not sufficient to induce the downregulation of Cdc25.

We hypothesized that Cdc25 could be a target (direct or indirect) of the pheromone MAPK cascade, which induces its degradation only when Cdc25 is retained at the cytoplasm and the activation of the pheromone cascade reaches some threshold level (as expected in the presence of high amounts of synthetic pheromone or expression of the activated *fuz7$^{DD}$* allele). To uncover the involvement of the MAPK cascade in the downregulation of Cdc25, we constructed a strain simultaneously expressing the *fuz7$^{DD}$* allele and *pcl12* under the *crg1* promoter. In this way, we can induce cell cycle arrest, as well as the retention of Cdc25 in the cytoplasm (as a consequence of the ectopic expression of *pcl12*), even in conditions that potentially disable the MAPK cascade, such as the disruption of the *kpp2* gene. In agreement with our hypothesis, we found that under conditions of cell cycle arrest and the retention of Cdc25 in the cytoplasm, Kpp2 was required for the decrease in Cdc25 levels upon *fuz7$^{DD}$* expression (*Figure 7C*).

Since we considered the possibility that Cdc25 was a direct target of Kpp2, we tried to detect physical interactions between Kpp2 and Cdc25. We used a strain that expressed both the *fuz7$^{DD}$* allele and *pcl12* and carried a kinase-dead allele of Kpp2 fused to GFP (Kpp2$^{K50R}$) (*Müller et al., 2003*), as well as an HA-tagged Cdc25 allele. Using GFP-trap beads, we were able to detect interactions between Kpp2 and Cdc25, which were dependent on the expression of the *fuz7$^{DD}$* allele (i.e., high levels of MAPK signaling) and the presence of a wild-type Kap123 allele (i.e., retainment of Cdc25 in the cytoplasm) (*Figure 7D*).

Taken together, our results were consistent with the idea that activation of the pheromone cascade resulted in the negative regulation of Cdc25 levels by a two-step mechanism: First, Cdc25 is retained in the cytoplasm as a result of the formation of the Pcl12-Crk1 complex, and then Cdc25 is potentially phosphorylated by the Kpp2 MAPK, which probably promoted its degradation.

Compared to other fungal Cdc25-like phosphatases, *U. maydis* Cdc25 carries an unusually long N-terminal extension of approximately 270 amino acids with no similarity to orthologues in the database (*Figure 7E*). Previous attempts to ascribe a role to this domain were elusive because its deletion did not appear to affect the Cdc25 function in *U. maydis* growing under axenic conditions (*Sgarlata and Pérez-Martín, 2005a*), although it affected the sensitivity of *U. maydis* cells to cell wall stressors (*Carbó and Pérez-Martín, 2010*). We found that the removal of this N-terminal extension resulted in the stabilization of Cdc25 upon expression of the *fuz7$^{DD}$* allele (*Figure 7F*). Based on sequence, this region contains at least three putative MAPK phosphorylation sites (Ser$^7$, Ser$^{23}$ and Ser$^{239}$, *Figure 5E*). With support from the observed physical interactions, we considered the possibility that Kpp2 phosphorylates Cdc25 in this region, promoting its degradation as described in mammalian Cdc25 in response to activation of the p38 MAPK (*Uchida et al., 2009*). To investigate this possibility, the three candidate Ser residues were mutated to Ala (*cdc25$^{AAA}$* allele). Encouragingly, we found that the triple mutant was resistant to the downregulation observed in the wild-type allele upon expression of the *fuz7$^{DD}$* allele (*Figure 7G*). Moreover, in accordance with our model explaining how the cell cycle is arrested in response to pheromone, neither the deletion of the N-terminal extension nor the alanine mutations affected the ability of cells undergo cell cycle arrest (*Figure 7H*) as well as the retention of Cdc25 at cytoplasm (*Figure 7I*).

In summary, these data strongly suggested that the MAPK Kpp2 targets and downregulates the phosphatase Cdc25, most likely promoting its degradation.

## The pheromone-dependent decrease of Cdc25 levels is required for virulence

We analyzed the capacity of sexually compatible strains (*a1 b1* and *a2 b2* mating type) carrying the *cdc25$^{AAA}$* allele to infect plants. Strikingly, we found that the presence of this mutant allele strongly affected the ability of fungal cells to produce disease to corn plants (*Figure 8A*). The defect in virulence could be attributed to problems related to the mating process itself or it might be related to

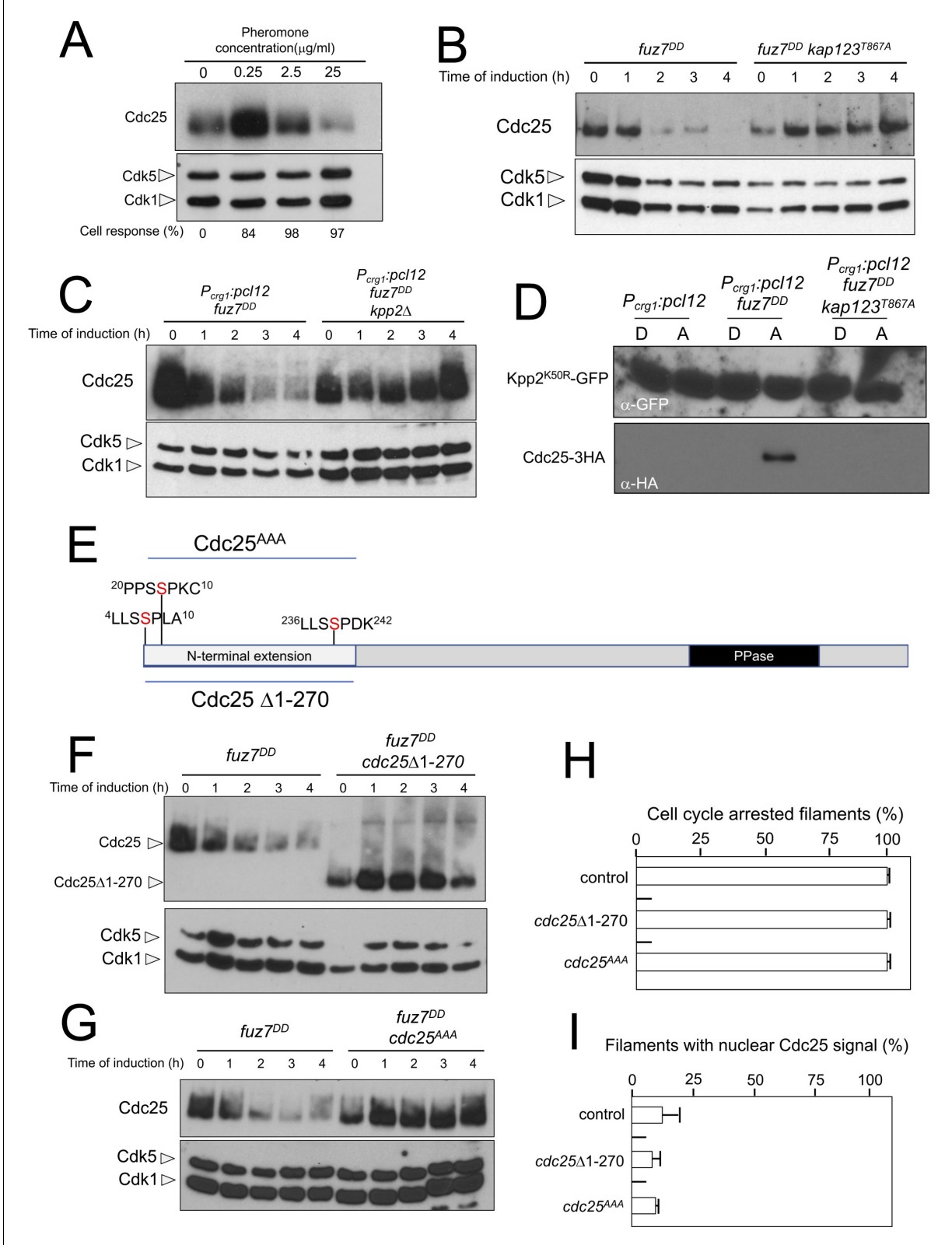

**Figure 7.** Cdc25 is negatively regulated by Kpp2, the pheromone MAPK. (**A**) High levels of pheromone resulted in a decrease in Cdc25 levels. Western blot from extracts obtained from *a1* mating-type cells carrying a HA-tagged Cdc25 endogenous allele that were incubated for 6 hr in the presence of the indicated synthetic a2 pheromone concentrations in CMD medium. Similar amount of protein extracts was separated by SDS-PAGE. Immunoblots were incubated with an antibody against HA. As loading control, we used the Cdk1 protein, which can be detected using anti-PSTAIRE (which

*Figure 7 continued on next page*

*Figure 7 continued*

recognizes both Cdk1 and Cdk5). Cell response refers to percentage of cells showing conjugation tube in each culture. (B) Inability to retain Cdc25 at the cytoplasm resulted in stabilization of Cdc25 levels in response to expression of the *fuz7^DD* allele. Western blot analysis to show the level of Cdc25 (upper blot) in cells growing in inducing conditions (CMA) for the expression of *fuz7^DD* allele during the indicated time. Levels of Cdk1 were used as loading control (bottom blot). (C) The MAPK Kpp2 is required for down-regulation of Cdc25 levels. Strains expressing at the same time *fuz7^DD* and *pcl12* and carrying or not a loss of function allele of *kpp2*, were grown during the indicated time in inducing conditions (CMA). Extracts were analyzed by Western blot to detect Cdc25-3HA protein levels (upper blot) and the levels of Cdk1, which were used as loading control (bottom blot). (D) Kpp2 is able to interact with Cdc25 upon activation of the pheromone cascade. Soluble extracts from strains carrying Cdc25-3HA and a Kpp2^KD-GFP tagged in their corresponding endogenous loci, and carrying ectopic copies of the indicated genes under the control of *crg1* promoter, as well as additional mutations (also indicated) were incubated with GFP-trap beads and the immunoprecipitates submitted to Western blot with anti-HA (Cdc25) and anti-GFP (Kpp2^KD) antibodies in succession. Cells were grown in inducing conditions (CMA, (A) or repressive conditions (CMD, (D) for *crg1* promoter during 6 hr. (E) Scheme of *U. maydis* Cdc25, showing the N-terminal specific extension, and the postulated MAPK phosphorylation sites, with the Serine residue (in red) exchanged to Alanine in the mutant allele *cdc25^AAA*. (F) The N-terminal extension of Cdc25 determines its down-regulation in response to expression of the *fuz7^DD* allele. Western blot analysis to show the level of Cdc25 or Cdc25 Δ1–270 (upper blot) upon expression of *fuz7^DD* allele in cells growing in inducing conditions (CMA) for the indicated time. Levels of Cdk1 were used as loading control (bottom blot). (G) Putative MAPK phosphorylation sites were involved in the down-regulation of Cdc25 in response to expression of the *fuz7^DD* allele. Western blot analysis to show the level of Cdc25 or Cdc25^AAA (upper blot) upon expression of *fuz7^DD* allele in cells growing in inducing conditions (CMA) for the indicated time. Levels of Cdk1 were used as loading control (bottom blot). (H) Cell cycle arrest in response to expression of the *fuz7^DD* allele is unaffected by the absence of down-regulation of Cdc25. *fuz7^DD*-expressing strains carrying the indicated *cdc25* alleles, which also carried the *NLS-GFP* transgene, were incubated in inducing conditions (CMA) for 6 hr. Filaments from each culture were counted and sorted as carrying 1 (cell cycle arrested) or more than one nucleus (not arrested). The graph shows the result from three independent experiments, counting more than 100 filaments each (*Figure 7—source data 1*). Means and SDs are shown. (I) Quantification of number of filaments showing GFP fluorescence associated with the nucleus in control and strains carrying the indicated *cdc25* alleles, expressing *pcl12* and incubated for 6 hr in inducing conditions (CMA). The graph shows the result from three independent experiments, counting 50 filaments each (*Figure 7—source data 2*). Means and SDs are shown.

DOI: https://doi.org/10.7554/eLife.48943.032

The following source data is available for figure 7:

**Source data 1.** Data for *Figure 7H*.
DOI: https://doi.org/10.7554/eLife.48943.033
**Source data 2.** Data for *Figure 7I*.
DOI: https://doi.org/10.7554/eLife.48943.034

subsequent steps, such as the ability of the fungus to proliferate within the plant, for example. To address this issue, we took advantage of the solopathogenic strain SG200, which is a haploid strain that carries the genetic information from the two different mating types and, as a consequence, does not require cell fusion to produce the infective hypha (*Bölker et al., 1995*). SG200-derived cells carrying the *cdc25^AAA* allele were as virulent as the control strain (*Figure 8A*), suggesting a problem in mating associated with the presence of the *cdc25^AAA* allele. Moreover, a *cdc25^AAA* mutant cross showed a weak fuzzy phenotype in charcoal plates, in comparison to wild-type control, indicating some kind of problem with the formation of the dikaryotic filament (*Figure 8B*). We wondered whether the impaired fuzzy phenotype in the mutant crosses was caused by defects in cell fusion. To address that question, we used the P_dik6-NLS-GFP reporter described above. Since this reporter will be only active in the presence of a functional b-factor, it can be used as an indirect readout of cell fusion between compatible sexual partners, observing the appearance of nuclear GFP fluorescence in the cell mixture resulting from cross. We crossed mutant haploid cells carrying the *dik6* reporter, and we were able to observe filaments carrying nuclear GFP signal, indicating that cell fusion was not affected. Interestingly, these filaments were multinucleated (*Figure 8C*).

In contrast to the multinucleated filaments found from crosses involving the Kap123^T867A mutant, which seemed not to be able to bypass the G2 barrier imposed by the b-factor (we never found more than four nuclei per filament, *Figure 6D*), a significant proportion of the filaments observed in *cdc25^AAA* crosses showed more than five nuclei (*Figure 8D*). Since cells carrying the *cdc25^AAA* allele were able to be arrested in response to activation of the pheromone cascade (*Figure 7H*), these observations prompted us to think that the presence of the *cdc25^AAA* allele could affect the G2 cell cycle arrest activated in the presence of a functional b-factor. However, we observed that the presence of the *cdc25^AAA* allele did not affect the ability of AB33 cells (see *Figure 6—figure supplement 1* for a description of this strain) to undergo cell cycle arrest upon the expression of compatible b proteins (*Figure 8E*).

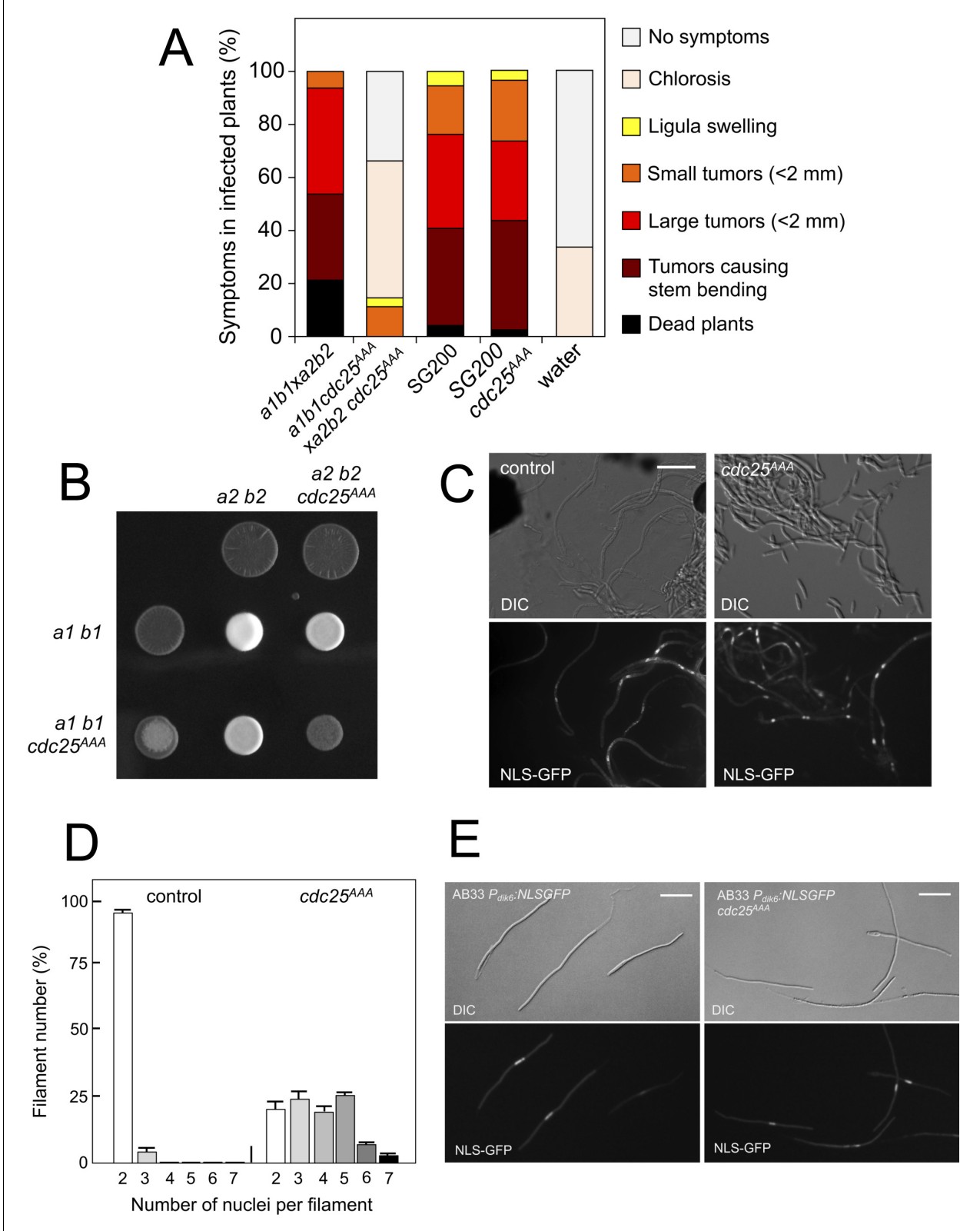

**Figure 8.** Pheromone-dependent down-regulation of Cdc25 is required for full virulence. (**A**) $cdc25^{AAA}$ mutant strains are affected in the ability to infect plants. Disease symptoms caused by crosses of wild-type and $cdc25^{AAA}$ mutant strains. Strikingly, solopathogenic SG200-derived strains were not affected. The symptoms were scored 14 days after infection. Two independent experiments were carried out and the average values are expressed as percentage of the total number of infected plants (n: 30 plants in each experiment) (**Figure 8—source data 1**). (**B**) $cdc25^{AAA}$ mutant is affected in the

*Figure 8 continued on next page*

*Figure 8 continued*

formation of dikaryotic hyphae. Crosses of control as well as *cdc25^AAA* mutant strains carrying compatible mating types (*a1 b1* and *a2 b2*) in charcoal-containing agar plates. Positive fuzzy phenotype can be detected as a white-appearance mycelial growth. Note that mutant combinations were affected in the ability to produce fuzzy phenotype. Plates were incubated at 22°C for two days. (B) *cdc25^AAA* mutant cells are able to fuse. Crosses from compatible strains (wild-type or *cdc25^AAA* mutant) carrying a NLS-GFP fusion under control of the b-factor-dependent *dik6* promoter were scrapped from agar surface, mounted on microscopy slides and epifluorescence was observed. Representative images show DIC and fluorescence in GFP channel. Bar: 20 μm. (C) *cdc25^AAA* mutant filaments were affected in nuclear number. Graph shows the quantification of the nuclear content of filaments obtained from charcoal plates. The graph shows the result from two independent experiments, counting more than 50 filaments each (*Figure 8—source data 2*). Means and SDs are shown. (E) b-induced cell cycle arrest de novo is not affected by the presence of *cdc25^AAA* mutant allele. Representative images of cultures from AB33-derived strains that were incubated in inducing conditions (minimal medium with nitrate) for 8 hr. Bar: 20 μm.

DOI: https://doi.org/10.7554/eLife.48943.035

The following source data is available for figure 8:

**Source data 1.** Data for *Figure 8A*.
DOI: https://doi.org/10.7554/eLife.48943.036
**Source data 2.** Data for *Figure 8D*.
DOI: https://doi.org/10.7554/eLife.48943.037

In summary, we observed that the absence of downregulation of Cdc25 levels in response to pheromone does not affect neither pheromone-induced cell cycle arrest nor de novo (independent of previous pheromone-induced cell cycle arrest) b-induced cell cycle arrest. However, the absence of downregulation of Cdc25 levels seems to affect the cell cycle arrest in b-dependent filaments originating from crosses (i.e., resulting from a mating process). These unexpected results can be explained keeping in mind that, in *U. maydis*, during a regular cell cycle Cdc25 accumulates at G2 phase; and, that in contrast to pheromone-dependent cell cycle arrest, the b-dependent cell cycle arrest can be bypassed by high levels of Cdc25 (*Mielnichuk et al., 2009*). We believe that during the pheromone-dependent G2 cell cycle arrest, Cdc25 is accumulating in the cytoplasm of conjugation tubes, but that as the compatible conjugation tubes are getting close (and therefore encountering higher pheromone concentrations) the subsequent increase in the pheromone-cascade signaling, promotes a decrease in the Cdc25 levels that allows a proper b-dependent cell cycle arrest in the filament once the cells were fused. Inability to achieve this decrease (as the case of stable Cdc25 alleles) could result in impaired b-dependent cell cycle arrest and thereby defects in pathogenicity.

## Discussion

Smut fungi are a widespread group of plant pathogens whose sexual development is coupled with virulence. All smut species investigated so far have in common the need to undergo a successful mating reaction to form an infective dikaryotic filament before being able to infect their host plant (*Bakkeren et al., 2008*). In the most studied member of this group, *U. maydis*, it has been described strong connections between the regulation of the cell cycle and the ability to infect plants: During the steps previous to plant infection, it is required a sustained G2 cell cycle arrest (*Perez-Martin, 2012*). The cell cycle arrest of the infective filament on plant surface observed in *U. maydis* seems to be more general, and it is also present in rust fungi like *Uromyces phaseoli* (*Heath and Heath, 1978*).

The connections between cell cycle regulation and the virulence in plant fungal pathogens have been studied in detail in other systems, in addition to *U. maydis*, such as *Magnaporthe oryzae* and *Colletotrichum orbiculare*. In these systems, the infection depends on the formation of appressorium, a structure required for plant penetration. The appressoria from these fungi use a turgor-driven mechanical process to breach the plant cuticle. The functionality of this class of appressorium implies the increase of internal turgor pressure, which is linked to an elaborated program that includes from metabolic reprograming to morphological changes to produce a clearly defined structure with a thick, multilayered and highly melanized cell wall (*Ryder and Talbot, 2015*). The various steps required for the formation of appressorium in these phytopathogenic fungi are linked to the different cell cycle phases. There is a G1/S control to induce the formation of appressorium, first described in *C. orbiculare* (*Fukada and Kubo, 2015*), but also operating in *M. oryzae* (*Fukada et al.,*

*2019*); and there is a S-phase checkpoint required for appressorium differentiation in *M. oryzae* (*Osés-Ruiz et al., 2017*; *Osés-Ruiz and Talbot, 2017*).

In contrast to appressoria from these phytopathogenic fungi, appressoria of *U. maydis* is not dependent on turgor pressure to penetrate the plant tissue. The appressorium directs the localized secretion of enzymes that weakens the plant cuticle, allowing the plant tissue penetration. For that reason, the morphogenesis process is less complex and appressoria are unmelanized, rather small swellings of the hyphal tip that form penetration structures that are less constricted (*Snetselaar and Mims, 1993*). However, in spite of this apparent lack of complexity, there is also a clear connection with cell cycle regulation. G2 cell cycle arrest is mandatory to promote the differentiation of the appressorium, and the impairment in cell cycle arrest resulted in a lack of virulence (*Castanheira et al., 2014*). This obligation for cell cycle arrest at G2 phase is most likely related to the fact that mitosis and the morphogenetic program responsible for appressorium formation compete for the same cytoskeletal components. Because of this competition, it makes sense that cellular controls exist to force these two processes to be incompatible and therefore to prevent infective filaments from entering mitosis (i.e., to arrest the progression of the cell cycle at G2) (*Pérez-Martín et al., 2016*).

G2 cell cycle arrest is established before mating and is maintained once the dikaryotic infective filament is formed, until the fungus enters the plant. During this period, cell cycle arrest is sustained by two distinct regulatory networks: First, by the pheromone-recognition cascade in each mating partner and second, once cytoplasmic fusion occurs, by the presence of a homeodomain transcriptional regulator called b-factor (which, among other targets, represses the *a*-locus, encoding the pheromone and receptors; *Urban et al., 1996*). Our results showed that the two regulatory networks recruit both shared and specific elements to induce and sustain cell cycle arrest (*Figure 9*). The G2/M transition in *U. maydis* is controlled by the level of inhibitory phosphorylation of the Cdk1-Clb2 complex, and both regulatory networks rely on an increase in Cdk1 inhibitory phosphorylation to arrest the cell cycle. To achieve this increase, both regulatory networks upregulate the kinase responsible for this inhibitory phosphorylation, the Wee1 kinase. This upregulation is indirect and is mediated by the transcriptional repression of *hsl1*, which encodes a kinase that represses Wee1 activity. However, the repression of *hsl1* is not enough to arrest the cell cycle at G2: loss-of-function mutants in Hsl1 are able to progress through the cell cycle, although they have an extended G2 phase (*Castanheira et al., 2014*). To arrest the cell cycle, a second element is required, namely the downregulation of Cdc25, the phosphatase that removes the Cdk1 inhibitory phosphorylation. Is at this step where the two regulatory networks differ. The pheromone cascade promotes the formation of a kinase-cyclin complex, Crk1-Pcl12, which phosphorylates and inhibits the β-importin Kap123, which is required for the nuclear translocation of Cdc25. The b-factor, however, activates the cascade responsible for the DNA damage response (DDR, composed of Atr1 and Chk1 kinases), resulting in phosphorylation of Cdc25, which promotes its cytoplasmic retention via 14-3-3 proteins (*de Sena-Tomás et al., 2011*; *Mielnichuk and Pérez-Martín, 2008*; *Mielnichuk et al., 2009*). This downregulation of Cdc25 also provides a time window to load a third element (not present in pheromone-induced cell cycle arrest) consisting of the transcriptional repression (via an intermediate regulator called Biz1) of *clb1*, which encodes a second b-type cyclin that is also required for G2/M transition (*Flor-Parra et al., 2006*).

The reasons for using distinct mechanisms to retain Cdc25 in the cytoplasm are probably related to the degree of reversibility of each developmental step during the infection process. The pheromone recognition by each mating partner has to culminate with the fusion of the respective conjugation tubes. If for some reason this step does not occur, the unmated cells must be able to return to the previous vegetative status. In other words, the pheromone cell cycle arrest should be reversible if the stimulus (pheromone) disappears and mating was not successful. A control mediated by the phosphorylation of importin could probably be easily reversed by some phosphatase (specific or not) once the signaling through the pheromone cascade is abrogated. The formation of the dikaryotic filament, however, is a more terminal decision in the sense that once the mating partners fuse their respective cytoplasms, if they are compatible at the *b*-locus, the infective filament is committed to infecting the plant. The presence of two independent cell cycle brakes, provided by the retention of Cdc25 at cytoplasm by 14-3-3 proteins and by the transcriptional repression of *clb1* could make this step less reversible.

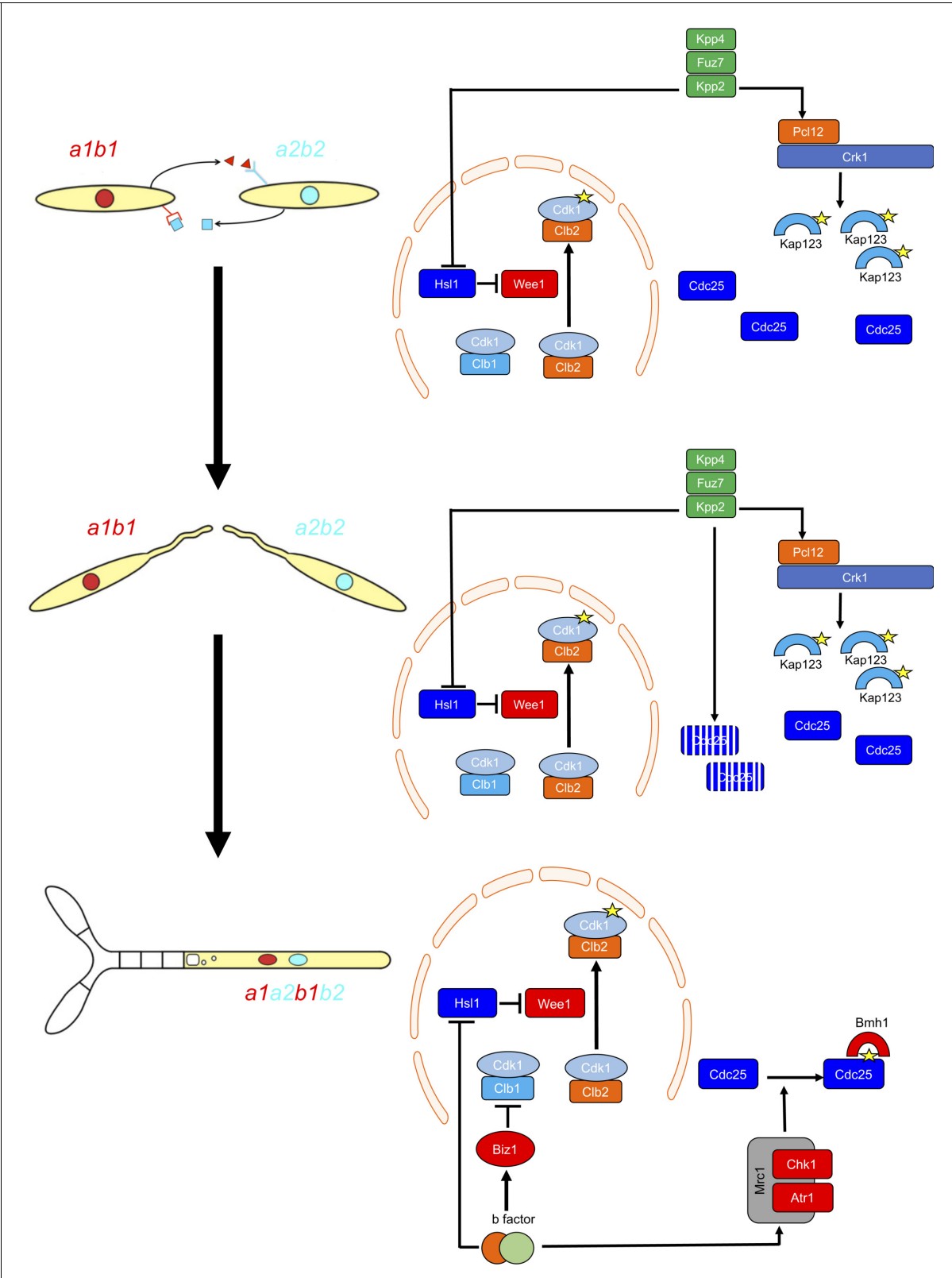

**Figure 9.** Two regulatory networks recruit both shared and specific elements to induce and sustain a G2 cell cycle arrest during the production of the infective structures in *U. maydis*. Scheme showing the distinct steps at cell cycle level occurring during the formation of the infective filament of *U. maydis*. Arrows and bars denote positive and negative interactions. Yellow stars denote phosphorylation. For details, see discussion section.

DOI: https://doi.org/10.7554/eLife.48943.038

Our results also reveal the importance of the transition from pheromone-dependent to b-dependent cell cycle arrest once the respective partners fuse. Although the downregulation of Cdc25 activity consists of the retention of Cdc25 in the cytoplasm in both regulatory networks, there is one relevant difference: while the inhibition of nuclear translocation (pheromone-dependent) cannot be bypassed by high concentrations of Cdc25, the retention of Cdc25 by 14-3-3 (b-dependent) is sensitive to the amount of Cdc25 (probably because of saturation of the available retention proteins). This difference gains importance because Cdc25 most likely accumulates in the cytoplasm of conjugation tubes during G2 cell cycle arrest. Once the respective cytoplasms fuse, if the amount of Cdc25 has not been previously tuned, the 14-3-3 retention system will probably be overwhelmed, and the time window required for the establishment of permanent G2 cell cycle arrest will not be provided. For that reason, it seems logical that once the pheromone cascade reaches some signaling threshold (probably reflecting a close encounter between compatible conjugation tubes), it triggers the downregulation of Cdc25 to levels that are most likely easily assimilable by the 14-3-3 proteins. The importance of adjusting the levels of Cdc25 is reflected by the impaired ability of cells carrying the $cdc25^{AAA}$ allele to infect plants.

Central to the pheromone-induced cell cycle arrest in *U. maydis* was the complex formed by the kinase Crk1 and the cyclin Pcl12. Both proteins were previously reported to be required for the proper morphogenesis of the infective filament in this fungus. Interestingly, in this function, they were part of distinct complexes or regulatory networks: Pcl12 complexed with the cyclin-dependent kinase Cdk5 (*Flor-Parra et al., 2007*), while Crk1 was activated in a similar way to MAPK via T-loop phosphorylation by the upstream MAPK kinase (*Garrido et al., 2004*). In this work, we showed that Pcl12 and Crk1 seem to form a complex with specific roles in cell cycle arrest upon pheromone recognition and that neither Cdk5 nor MAPK cascade phosphorylation are involved. How Pcl12 controls the activity of Crk1 in this process is beyond the scope of this work. However, we can anticipate some possibilities. Crk1 belongs to the family of RD kinases, which have a conserved arginine residue preceding an aspartate residue in the catalytic loop (*Johnson et al., 1996*). For this reason, the members of this family require an activation segment (T-loop) that undergoes a conformational change to fold into an active catalytic site. This conformational change can be achieved either by the phosphorylation of specific residues (such as TxY in MAP kinases) or by interaction with other proteins (such as cyclins in cyclin-dependent kinases) (*Nolen et al., 2004*). Crk1 carries a TEY signature at its T-loop, similar to a MAP kinase, but it is worth mentioning that it was cloned from a genetic screening devoted to uncovering CDK-like proteins in *U. maydis* and that the predicted three-dimensional folding of the catalytic domain of Crk1 fits the structure of Cdk2 (*Garrido and Pérez-Martín, 2003*). We propose that Crk1 can be activated either by T-loop phosphorylation (through the pheromone MAPK cascade) or by interaction with Pcl12 cyclin. The activation of Crk1 in the absence of Pcl12, which is dependent on T-loop phosphorylation, resulted in the promotion of polar growth without affecting cell cycle regulation (*Garrido et al., 2004*). However, cell cycle arrest by Crk1 was dependent on Pcl12 cyclin but independent of T-loop phosphorylation. The use of proteins unrelated to cyclins to activate CDKs is well described, including the so-called K-cyclins from viruses, p35 as a coactivator of mammalian Cdk5 and the family of RINGO/Speedy proteins (*Nebreda, 2006*). However, the use of a cyclin to activate a noncanonical cyclin-dependent kinase has not, to the best of our knowledge, been previously described. The importance of this alternative activation mechanism can be appreciated by considering that Crk1 belongs to a widely conserved family of fungal kinases, the founding member of which is Ime2, a central regulator of meiosis in *S. cerevisiae*. The Ime2-related kinases exhibit amazing variety in controlling sexual developmental programs in fungi, although the targets and physiological inputs seem to be very species-specific (*Irniger, 2011*). Alternative mechanisms of activation could help to explain this plethora of roles.

# Materials and methods

## Key resources table

| Reagent type (species) or resource | Designation | Source or reference | Identifiers | Additional information |
|---|---|---|---|---|
| *Ustilago maydis* | Strains used in this study are listed in *Supplementary file 1* | Perez-Martin lab | N/A | |
| *E. coli* strain | DH5α | CGSC | 12384 | |
| Antibody | Anti-PSTAIRE (Rabbit polyclonal) | Santa Cruz Biotechnology, Inc | Sc-53 | RRID:AB_2074908 |
| Antibody | Anti-phospho-Cdc2 (Tyr15) (10A11, Rabbit monoclonal) | Cell Signaling | 4539 | RRID:AB_560953 |
| Antibody | Anti-HA, High affinity (3F10, Rat monoclonal) | Roche | 1 867 423 | RRID:AB_390919 |
| Antibody | Anti-HA-HRP conjugate, High affinity (3F10, Rat monoclonal) | Roche | 12 013 819 001 | |
| Antibody | Anti-c-Myc-HRP conjugate (9E10, mouse monoclonal) | Roche | 1 814 150 | |
| Antibody | Anti-GFP, Living Colors (JL-8, mouse monoclonal) | Clontech | 632380 | RRID:AB_2314359 |
| Antibody | Anti-rabbit IgG-HRP conjugate (Donkey polyclonal) | Amersham Biosciences | NA934 | RRID:AB_772206 |
| Antibody | Anti-rat IgG-HRP conjugate (Mouse, monoclonal) | SIGMA | R7636 | RRID:AB_1840005 |
| Antibody | Anti-mouse IgG-HRP conjugate (Goat polyclonal) | SIGMA | A0168 | RRID:AB_257867 |
| Chemical compound, drug | Hygromycin B | Roche | 834 555 | |
| Chemical compound, drug | clonNAT (nourseothricin) | Werner BioAgents | CAS#96736-11-7 | |
| Chemical compound, drug | G418 | Formedium | G418-1 | |
| Chemical compound, drug | Carboxine | SIGMA | 45371 | |
| Chemical compound, drug | Phleomycin | InvivoGen | ant-ph-1 | |
| Chemical compound, drug | Roche Protease Inhibitor Cocktail | Roche | 11-697-498-001 | |
| Chemical compound, drug | PhosSTOP | Roche | 04-906-837-001 | |
| Chemical compound, drug | Synthetic *U. maydis* a2 pheromone | Proteomic Services from National Center of Biotechnology, CSIC, Madrid | N/A | |
| Commercial assay or kit | High Pure RNA isolation kit | Roche | 11828665001 | |
| Commercial assay or kit | GFP-Trap_MA | Chromotek | gtma-400 | |

*Continued on next page*

*Continued*

| Reagent type (species) or resource | Designation | Source or reference | Identifiers | Additional information |
|---|---|---|---|---|
| Commercial assay or kit | IgG coupled to Dynabeads (M-270 Epoxy) | Thermo Fisher | 14304 | |
| Commercial assay or kit | High-Capacity cDNA Reverse Transcription Kit | Applied Biosystems | 4368814 | |
| Commercial assay or kit | SsoAdvanced SYBR Green supermix | BioRad | 172–5260 | |
| Commercial assay or kit | Supersignal West Femto Maximum Sensitivity Substrate | Thermo Scientific | 34095 | |
| Commercial assay or kit | Clarity Western ECL Substrate | Bio-Rad | 170–5061 | |
| Commercial assay or kit | Mini-Protean TGX gels | Bio-Rad | 456–1095 | |
| Software, algorithm | Image J | NIH | https://imagej.nih.gov/ij/ | |
| Software, algorithm | Photoshop | Adobe | https://www.adobe.com | |

## *U. maydis* growth conditions

*Ustilago maydis* strains are derived from FB1 and FB2 genetic backgrounds (*Banuett and Herskowitz, 1989*) and are listed in *Supplementary file 1*. Cells were grown in rich medium (YPD), complete medium (CMD or CMA) or minimal medium (MMD) (*Holliday, 1974*). Controlled expression of genes under the *crg1* and *nar1* promoters was performed as described previously (*Brachmann et al., 2001*). FACS analyses were described previously (*García-Muse et al., 2003*).

*U. maydis* strains used in the presented experiments are as follows:

*Figure 1* (A) FB1, FB1Pcrg:fuz7DD; (B) UMS2; (C) FB1, FB1Pcrg:fuz7DD; (D) UMS2, UMS3, UMS5, UMS15; (E) UMPB12, UMPB15, UMS195, UMPB6, UMS200, UMPB14; (F) UMS1, UMPB109.
*Figure 2* (A) UMS2, UMS125, UMS136; (B) UMPB16, UMPB18, UMPB25; (C) UMS176, UMS177, UMS179, UMS243, UMS244, UMS245, UMS246, UMS2, UMS125, UMS184, UMS185; (E) UMS176, UMS177, UMS179; (F) UMS2, UMS184, UMS185.
*Figure 3* (A) UMPB111, UMPB112; (C) UMPB130, UMPB131, UMPB132, UMPB135; (D AND E): UMPB114, UMPB115; (F) UMPB114, UMPB116; (G) UMPB 114, UMPB115, UMPB120, UMPB121, UMPB122; (H) UMPB114, UMPB116, UMPB125, UMPB126, UMPB127.
*Figure 4* (B) UMPB115, UMPB143; (C) UMPB116, UMPB146; (D) UMPB148, UMPB149, UMPB150, UMPB151; (F) UMPB158; (H) UMPB131, UMPB156.
*Figure 5* UMPB160, UMPB161.
*Figure 6* (A) UMPB170, UMPB171, UMPB172, UMPB173; (B–D) UMPB171, UMPB173, UMPB174, UMPB175.
*Figure 7* (A) UMS195; (B) UMPB16, UMPB179; (C) UMPB162, UMPB163; (D) UMPB164, UMPB165, UMPB166; (F) UMPB16, UMPB168; (G) UMPB16, UMPB169; (H) UMS2, UMPB180, UMPB181; (I) UMPB111, UMP320, UMP321.
*Figure 8* (A) FB1, FB2, UMPB186, UMPB187, SG200, UMPB192; (B–D) UMPB188, FB2, UMPB189, UMPB187; (E) UMP121, UMPB190.
*Figure 1—figure supplement 1* (D) FB1Pcrgfuz7DD; (E) FB1Pcrgfuz7DD, UMS26, UMS28.
*Figure 1—figure supplement 2* (A) FB1, FB1Pcrgfuz7DD; (C and D) UMS2, UMS3, UMS5; (F) UMS2, UMS15.
*Figure 1—figure supplement 3* (A) FB1, FB1Pcrgfuz7DD; (B) UMS2, UMS51, UMS73, UMS175; (C) UMS2, UMS73.
*Figure 1—figure supplement 4* (B–D) UMS2, UMPB109.
*Figure 2—figure supplement 1* (A) UMS2, UMS203; (C) UMPB16, UMPB62.
*Figure 2—figure supplement 2* FB1Pcrgfuz7DD, UMPB33
*Figure 2—figure supplement 3* UMS176, UMS243, UMS244, UMS245, UMS246.
*Figure 2—figure supplement 4* (A) UMS2, UMS125, UMS136, UMS144.

## *U maydis* strain generation

To construct the different strains, transformation of *U. maydis* protoplasts with the desired constructions was performed as described previously (*Tsukuda et al., 1988*). Integration of the corresponding construction into the corresponding loci was verified in each case by diagnostic PCR and subsequent Southern blot analysis or RT-PCR analysis of transcripts depending on the type of integrated mutant allele. *U. maydis* DNA isolation was performed as previously described (*Tsukuda et al., 1988*).

## Plasmid construction

Plasmid pGEM-T easy (Promega) and pJET1.2 (Thermo Fisher) was used for subcloning and sequencing of genomic fragments generated by PCR.

Plasmids for C-terminus end tagging of endogenous loci from *clb2*, *cdc25*, *wee1*, and *kap123* with 3 HA or GFP as well as conditional alleles of *kap123* and *srp1* were performed using Golden Gate assembly, following published procedures (*Terfrüchte et al., 2014*). Mutant alleles (loss of function and allelic variants) of the following genes were already described: *crk1* (*Garrido and Pérez-Martín, 2003*; *Garrido et al., 2004*); *pcl12* (*Flor-Parra et al., 2007*); *kpp2*, *fuz7* and *prf1* (*Müller et al., 2003*); *cdk1* and *wee1* (*Sgarlata and Pérez-Martín, 2005b*); *cdc25* and *hsl1* (*Castanheira et al., 2014*); *NLS-GFP* (*Mielnichuk et al., 2009*); *cut11-cherry* (*Pérez-Martín, 2009*).

Kap123 serine or threonine to alanine mutant variants were constructed as cassettes carrying the desired mutation (associated with the appearance or loss of a restriction enzyme target for diagnostic purposes) to be inserted by homologous recombination into the endogenous locus. The cassette also allowed the insertion at the C-terminus of a 3 HA epitope as well as an antibiotic resistance marker. For *kap123*$^{T867A}$ allele used in infections and mating assays, a specific cassette without 3 HA epitope was constructed, associated with a resistance to hygromycin, flanked by FRT sites to remove the resistance cassette once the endogenous locus was mutated, following published procedures (*Khrunyk et al., 2010*). Cdc25 tagged at the N-terminus as well as the N-terminal Δ1–270 and *cdc25*$^{AAA}$ mutants, were constructed by inserting the corresponding mutant cassettes at the N-terminus associated with a resistance to hygromycin flanked by FRT sites to be removed upon transformation with a plasmid encoding a FLPase recombinase (*Khrunyk et al., 2010*).

Further details of the constructions explained above are available on request.

## RNA analysis

Total RNA was extracted with acidic phenol solution. After extraction, the RNA was cleaned using the High Pure RNA Isolation Kit (Roche Diagnostics GmbH). For RT- PCR, cDNA was synthesized using the High Capacity cDNA Reverse Transcription Kit (Applied Biosystems) employing 1 µg total RNA per sample. qRT-PCR was performed using the SsoAdvanced Universal SYBR Green Supermix (BioRad) in a CFX96 Real-Time PCR system (BioRad). Reaction conditions were as follows: 3 min 95˚C followed by 40 cycles of 10 s 95˚C/10 s 60˚C/30 s 72˚C.

## Cell lysates preparation and gel electrophoresis analyses

Protein extracts were performed using an adapted chloroacetic acid (TCA) method. Briefly, cells from 5- to 10 ml aliquots of cultures were harvested, and 1 ml of 20% TCA was added. The supernatant was removed after centrifugation, and the pellet was resuspended in 100 µl of 20% TCA and stored at −80˚C for 1 hr. Samples were thawed on ice, glass beads were added, and cells were broken using a FastPrep FP120 cell disrupter (BIO 101 ThermoSavant, Obiogene, Carlsbad, CA). The lysate was recovered by punching a hole on the bottom of the tube, and the glass beads were further washed with 200 µl of 5% TCA. Lysates were centrifuged at 1000 × *g* for 3 min, and the pellet was thoroughly resuspended in 100 µl of 2 × Laemmli buffer and 50 µl of 2 M Tris base. After boiling for 5 min, 10–20 µl were loaded in the gels.

For general purposes, TGX (4–20%) gels from BioRad were used, at constant 100 v running conditions. To detect the phosphorylated forms of Kap123, 8% acrylamide/0.1% bisacrylamide, pH 9.2 custom-made gels were used at constant 5mA running conditions.

Western blots were repeated from at least three independent experiments in each case.

## Protein–protein interaction analysis by Co-immunoprecipitation

To perform immunoprecipitations, crude protein extracts were prepared. Briefly, cells were harvested by centrifugation at 4°C and washed twice with ice-cold water. The cell pellet was resuspended in ice-cold HB buffer (25 mM MOPS pH 7.2, 15 mM MgCl$_2$, 15 mM EGTA, 1% Triton X-100, PhosSTOP [1 tablet per 10 mL], Roche protease inhibitor cocktail [1 tablet per 10 mL]) and cells were broken using a FastPrep FP120 cell disrupter. The lysate was recovered by punching a hole on the bottom of the tube, and the glass beads were further washed with ice-cold HB buffer, and the lysated was cleared by centrifugation (10 min/13000xg).

For co-immunoprecipitation analysis, approximately 3.5 mg of total protein extracts (1 ml) were incubated with 1 µg of the monoclonal antibody for 2 hr at 4°C and then prewashed G-protein coupled magnetic beads (50 µl) were added and incubated for 30 min at 4°C with agitation. For GFP trap, 50 µl GFP trap beads were mixed with 3.5 mg of total protein extracts (1 ml) for 2 hr at 4°C, with agitation. Immunoprecipitates were washed six times with 1 ml of HB buffer.

## Proteomic sample preparation and LC-MS analysis of peptides

Crude extracts were obtained and incubated with GFP-trap beads as explained above. After separation of the protein samples by SDS-PAGE and Coomassie Brilliant Blue R250 (Serva, 17525, Heidelberg, Germany) staining and distaining of the SDS-Gel was performed and each lane was cut into 10 pieces. A digest with trypsin NB sequencing grade (Serva, 37283.01, Heidelberg, Germany) was performed with each gel piece over night at 37°C. The eluates of the five upper and the five lower gel pieces were combined, suspended in sample buffer (98% H$_2$O, 2% acetonitrile and 0.1% formic acid) and analyzed by LC-MS.

Peptides generated with trypsin on-bead digestion were subjected to LC-MS analysis: 1.5 µL of each peptide sample were separated with nano-flow LC using an RSLCnano Ultimate 3000 system (Thermo Fisher Scientific). Peptides were loaded for 5 min with 0.07% trifluoroacetic acid on an Acclaim PepMap 100 pre-column (100 µm x 2 cm, C18, 3 µm, 100 Å; Thermo Fisher Scientific) with a flow rate of 20 µL/min. Separation of peptides was done with reverse phase chromatography on an Acclaim PepMap RSLC column (75 µm x 50 cm, C18, 3 µm, 100 Å; Thermo Fisher Scientific) at a flow rate of 300 nL/min. The solvent composition was gradually changed within a time period of 94 min from 96% solvent A (0.1% formic acid) and 4% solvent B (80% acetonitrile, 0.1% formic acid) to 10% solvent B within 2 min, to 30% solvent B within the following 58 min, to 45% solvent B within the next 22 min, and to 90% solvent B within the following 12 min. All solvents and acids had Optima LC/MS quality and were purchased from Thermo Fisher Scientific. Eluting peptides were on-line ionized with nano-electrospray (nESI) using the Nanospray Flex Ion Source (Thermo Scientific) at 1.5 kV (liquid junction) and on-line transferred into an Orbitrap VelosPro mass spectrometer (Thermo Fisher Scientific). Full scans were recorded in a mass range of 300 to 1650 m/z at a resolution of 30,000 followed by data-dependent top 15 CID fragmentation (dynamic exclusion enabled). LC-MS method programming and data acquisition was performed with the XCalibur 4.0 software (Thermo Fisher Scientific).

## Plant infections, mating assays

Pathogenic development of wild type and mutant strains was assayed by plant infections of the maize (*Zea mays*) variety Early Golden Bantam (Olds seeds) as described before (*Kämper et al., 2006*). For mating assays, strains were crossed on charcoal-containing complete medium plates and incubated at 22°C (*Holliday, 1974*).

## Microscopy

Images were obtained using a Nikon Eclipse 90i fluorescence microscope with a Hamamatsu Orca-ER camera driven by Metamorph (Universal Imaging, Downingtown, PA). Images were further processed with ImageJ software.

To determine the presence of Cdc25 at nucleus, merged (RGB) images from cells carrying a cut11-cherry allele (marking the nuclear membrane) and a GFP-Cdc25 allele were used. In each nucleus, a line was traced covering all nucleus diameter and also surrounding cytoplasm using the ImageJ software and the plot intensity for each channel was obtained (using the RGB line profile plugging from ImageJ). Those cells in which the signal inside the nucleus was above the cytoplasmic background were considered as positive for nuclear GFP. In the few cases of strong cytoplasmic background, the sorting or not of that particular nucleus was decided case by case, by looking the intensity of nuclear signal.

### Quantification and statistical analysis

To determine the statistical significance of differences a two-tailed Student $t$-test was used. $P$-Values were calculated with the GraphPad Prism 5.0 software.

## Acknowledgements

We thank Dr. K Heimel (Georg-August-University, Göttingen, Germany) for the help in sample preparation for LC/MS analysis, and members of the Pérez-Martín lab for discussion and critical reading of the manuscript. PB was supported by a Marie Curie ITN Grant (FUNGIBRAIN, FP7-PEOPLE-2013-ITN-607963). SC was supported by a Marie Curie ITN Grant (ARIADNE, PITN-GA-2009–237936). This work was supported by Grants from Spanish Government (BIO2014-55398-R and BIO2017-88938-R) to JPM and by Grants from the Deutsche Forschungsgemeinschaft (DFG) to GHB.

## Additional information

### Funding

| Funder | Grant reference number | Author |
|---|---|---|
| European Commission | Marie Curie ITN Grant (FUNGIBRAIN FP7-PEOPLE-2013-ITN-607963) | Paola Bardetti |
| European Commission | Marie Curie ITN Grant (ARIADNE PITN-GA-2009-237936) | Sónia Marisa Castanheira |
| Deutsche Forschungsgemeinschaft | | Gerhard H Braus |
| Spanish Government | BIO2014-55398-R | José Pérez-Martín |
| Spanish Government | BIO2017-88938-R | José Pérez-Martín |

The funders had no role in study design, data collection and interpretation, or the decision to submit the work for publication.

### Author contributions

Paola Bardetti, Sónia Marisa Castanheira, Investigation, Methodology, Writing—review and editing; Oliver Valerius, Resources, Writing—review and editing; Gerhard H Braus, Resources, Funding acquisition, Writing—review and editing; José Pérez-Martín, Conceptualization, Supervision, Funding acquisition, Investigation, Methodology, Writing—original draft, Project administration, Writing—review and editing

### Author ORCIDs

José Pérez-Martín  https://orcid.org/0000-0001-9849-7382

### Decision letter and Author response

Decision letter https://doi.org/10.7554/eLife.48943.046
Author response https://doi.org/10.7554/eLife.48943.047

# Additional files

## Supplementary files

• Supplementary file 1. *U. maydis* strains used in this study.
DOI: https://doi.org/10.7554/eLife.48943.039

• Transparent reporting form DOI: https://doi.org/10.7554/eLife.48943.040

## Data availability

All data generated or analysed during this study are included in the manuscript and supporting files. Source data files have been provided for Figures 1-8.

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
