## [Decision Letter]

Thank you for submitting your article "Cytoplasmic retention and degradation of a mitotic inducer enable the steps for plant infection by a pathogenic fungus" for consideration by *eLife*. Your article has been reviewed by Christian Hardtke as the Senior Editor, a Reviewing Editor, and two reviewers. The reviewers have opted to remain anonymous.

The reviewers have discussed the reviews with one another and the Reviewing Editor has drafted this decision to help you prepare a revised submission.

Both reviewers agree that your work is of interest. However, they are also of the opinion that your data do not yet unequivocally support your interpretation, especially the novelty aspect of temporarily separate G2 blocks that are achieved by diverse mechanisms. It is essential that you address these issues comprehensively in a revised version of your manuscript.

We also all agreed that your manuscript requires extensive re-writing to make your study more easily accessible and comprehensible for the average reader. In its current form, we all found the story difficult to follow.

Finally, please do your best to address, wherever possible within a reasonable time frame, the other comments in the reviews pasted below.

*Reviewer #1:*

In this manuscript, the authors provide a large amount of data addressing the mechanisms by which the corn smut Ustilago maydis arrests the cell cycle in G2 in response to pheromone during and after mating, which precedes plant colonization. Their main findings are (1) that activation of the pheromone-MAPK cascade leads to retention of the Cdc25 phosphatase in the cytosol due to phosphorylation of the Cdc25 importin likely by a complex formed between the cyclin Pcl12 and the CDK-like kinase Crk1, and (2) high-level activation of the pheromone-MAPK kinase leads to downregulation of Cdc25 protein levels due to phosphorylation of the N-terminal tail of Cdc25 likely by the MAPK Kpp2. Interestingly, preventing the cell cycle block by blocking importin phosphorylation does not prevent mating, nor plant pathogenesis. By contrast, preventing Cdc25 downregulation has no effect on pre-fusion cell cycle arrest, but impairs post-fusion cell cycle block and pathogenesis. The data is interpreted in terms of importance of this dual regulation during a cell fate change.

This is an interesting story, for which a lot of the data is solid, with genetics and biochemistry supporting the claims. There are, however, a few areas that need to be looked at to make it a strong paper for publication.

Experiments/data analysis:

The figures showing the localization of GFP-Cdc25 are not convincing (except Figure S10C). Providing both single channels and merge would help, maybe of single examples at higher magnification. This should be coupled with quantification of nuclear and cytosolic levels.

One important message from the paper is that Cdc25 is regulated by two distinct mechanisms: a cytosolic retention mechanism keeping pre-mating cells in G2, and a degradation mechanism important to keep post-mating cell in G2. From the data using mating cells, I am not fully convinced that this distinction in temporality can be made. First, the authors claim that mating appears to proceed normally in the kap123-T867A mutant, which fails to block the cell cycle in G2 when fuz7-DD or pcl12 are overexpressed. However, Figure 4A, showing Kap123 migration pattern in response to pheromone stimulation, is not convincingly showing the claimed mobility shift. The gel appears to be slightly smiley, but I don't see bands migrating slower. This lowers the confidence that Kap123 is in fact phosphorylated in these conditions. In addition, though kap123-T867A mutants form filaments that often have >2 nuclei in mating conditions (Figure 4D), it is not clear whether this results from a pre-mating failure in G2 block as claimed by the authors, or a post-mating block failure. In fact, when assessing the cdc25-AAA mutant, the authors perform exactly the same experiment (Figure 6C) and interpret the multi-nucleated defect as a post-fusion cycling issue. Imaging the filaments pre- and post-fusion in these conditions would address the issue.

Similarly, because Cdc25 levels decrease with increasing dosage of pheromone, the authors hypothesize that Cdc25 is a low-affinity substrate for the MAPK (subsection “The MAPK Kpp2 directs Cdc25 downregulation”), which is phosphorylated and destabilized only upon high pheromone signal. However, the authors do not test whether the cytosolic retention of Cdc25 is also pheromone-level dependent. It could equally be that Kap123 importin phosphorylation scales with pheromone concentration and that Cdc25 is retained more in the cytosol in high pheromone levels, which will lead to more Cdc25 phosphorylation and degradation given this only happens in this compartment. These issues need to be clarified.

It is unfortunate that the paper just falls short of demonstrating a direct substrate-kinase relationship between the importin and Crk1 and between Cdc25 and Kpp2, though it shows existence of complexes between the importin and Ltc12, and between Cdc25 and Kpp2.

Text changes:

The writing needs to be looked at. From the abstract it is not clear whether there are two distinct cell cycle arrests during distinct cycles, or two distinct mechanisms of arrest during the same cell cycle. It is only after reading the paper that I understood that the two arrests refer to two distinct mechanisms that occur during mating of haploid cells pre-fusion and in dikaryotic cells post-fusion, which together maintain the cell in the same G2 phase. The same goes in other parts of the manuscript. For instance, "the mutant filaments" (subsection “The pheromone-dependent decrease in the Cdc25 level enables b dependent cell cycle arrest.”) could be more clearly spelled out as dikaryotic post-fusion cells. There are also a number of grammatical mistakes.

In the Discussion section, I do not understand in what way repression of clb1 transcription by Biz1 would be irreversible. The discussion spends much time discussing these regulatory mechanisms, which are not the object of this paper, but does not discuss some pertinent questions raised by the data, for instance why Cdc25 levels do not drop upon Pcl12 over-expression but Pcl12 is required for the Cdc25 levels drop upon fuz7-DD expression. My guess is that Cdc25 downregulation can only happen in the cytosol and that Pcl12 is required for Cdc25 cytosolic retention, but this is not explicitly formulated.

*Reviewer #2:*

Cytoplasmic Retention and degradation of a mitotic inducer enable the steps for plant infection by a pathogenic fungus. Bardetti et al.

This study seeks to determine the mechanism of cell cycle control that leads to plant infection by the corn smut fungus Ustilago maydis. During infection, the fungus forms a dikaryon, which is the infective state. This requires synchrony between the cell cycles of the fusing haploid sporidia, so that subsequent development is co-ordinated. There are two discrete cell cycle arrests. The first is initiated upon pheromone recognition by each mating partner, while a second cell cycle arrest occurs after cell fusion leading to establishment of the infective dikaryotic filament. The significance of this study is that the authors propose that the cell cycle arrests occur by two distinct mechanisms. Furthermore, the first pheromone-induced cell cycle arrest results from inhibition of the nuclear import of the mitotic inducer Cdc25, based on inhibiting the function of the importin, Kap123. This is a novel mechanism, distinct from what is known in the model yeasts. Interestingly, the absence of the first G2 cell cycle arrest is dispensable for virulence, while the second cell cycle control point is essential. Linking the two control mechanisms is therefore necessary.

The paper is interesting because it reveals a new mechanistic insight into the control of virulence by a dikaryotic basidiomycete pathogen. This builds upon the pioneering work of the Kahmann lab, in particular (plus Kronstad, Banuett and others) in working out how pheromone signalling and self/non-self recognition via bE/bW mating type locus functions. The link to cell cycle control is, however, vital, but often overlooked, so this paper is potentially significant, building on some excellent previous work by lead author Perez-Martin. Furthermore, the work is carefully executed in the main. What really lets it down (but which I hope can be remedied) is the way the paper is written. The manuscript is quite impenetrable in parts and needs to be re-drafted to include a set of intermediate conclusions at each stage in which the work is presented. This is absolutely vital, because currently I think the number of people who will understand this work, let alone appreciate its significance, is currently very small.

The Abstract and Introduction section describe the conclusions in a relatively straightforward way, including the key finding that the pheromone-induced G2 arrest is a consequence of inhibition of nuclear transport of Cdc25 via Kap123. I think the key point that is missed in the Results section is the comparison with established mechanism by which b-factor induces G2 cell cycle arrest. I think the opening of the Results section should be re-written in a way that structures more tightly to the three effects by which b-factor mediates the sustained cell cycle arrest? This will help to explain the rationale for the study and the way in which the experiments were carried out. It will, for example, help to point out the differences in relation to the results of cdc25 transcriptional levels found in Figure 1—figure supplement 3B with protein levels reported in Figure 1E and Figure 1F, that are important to understand if the rest of the paper is to be understood.

Essential revisions:

1) In each section, the authors need to make a firm conclusion to help orient the reader. This is almost entirely missing, which makes the manuscript really difficult to follow. In the subsection” Activation of the pheromone cascade promotes G2 cell cycle arrest via inhibitory phosphorylation of Cdk1”, the rationale for using the fuz7DD mutant, for instance, is not fully explained and there is no conclusion provided to subsection “Activation of the pheromone cascade promotes G2 cell cycle arrest via inhibitory phosphorylation of Cdk1” at all.

2) The introduction of the subsection “Pheromone cascade-induced cell cycle arrest depends on an alternative cyclin interacting with a Cdk-like kinase” has no introduction and does not follow logically from the previous section. The rationale for this part of the paper is not introduced until later in this subsection. This needs to come sooner, and with a better explanation.

3) In Figure 1, I would prefer to see discrete data points rather than the bar charts shown in 1C and D.

4) The Cdc25 overexpression experiment (subsection “Activation of the pheromone cascade promotes G2 cell cycle arrest via inhibitory phosphorylation of Cdk1”) needs more emphasis, as it is critical for understanding why the nuclear localisation idea was followed. I think some of this data should be in Figure 1 and not in Figure 1—figure supplement 4.

5) Figure 2A- please add a micrograph of pcl12.

6) Figure 2B showing Cdk1 phosphorylation in FuzDD Δpcl12. The blot in 2B seems to show less total Cdk1 and Cdk5. Instead of having Figure 3—figure supplement 1, it might be better to more clearly to show CDK1 phosphorylation in Figure 2B. Could maybe authors clarify why they show Figure 3—figure supplement 1 vs. a blot of CDK1P?

7) Please clarify sentences subsection “Pheromone cascade-induced cell cycle arrest depends on an alternative cyclin interacting with a Cdk-like kinase” on the role of CDK and link with MAPK and Prf1 signaling. It is not clear what authors are trying to point out here and it is not clear that this work is already published.

8) In subsection “Pheromone cascade-induced cell cycle arrest depends on an alternative cyclin interacting with a Cdk-like kinase”, the authors suggest that Pcl12 and Crk1 interact to cause the cell cycle arrest. Can a physical interaction be demonstrated either by co-localization of these proteins in arrested vs. non-arrested cells or in vitro by Y2H to show if this is direct or indirect interaction. This statement really needs much better support as no evidence is provided.

9) In Figure 2E, a GFP-cdc25 control is needed to show difference between arrested vs. non-arrested cells.

10) Figure 2 we understand that Cdc25 degradation depends on Pcl12 and independent on crk1. But this conclusion reads diluted within the text. For example, the conclusion comes in the end of subsection “Pheromone cascade-induced cell cycle arrest is mediated by the phosphorylation of Cdc25 importin, Kap123” all together but maybe authors could slowly build up during the progression of the text this conclusion.

11) Subsection “Pheromone cascade-induced cell cycle arrest depends on an alternative cyclin interacting with a Cdk-like kinase”. Again, there is no conclusion made to this section at all, which means that the next (really important) part of the paper on the Kap123 protein is introduced without any real reasoning.

12) What happen to nuclear localization of Cdc25AAA? Has this been checked?

13) Have the authors expressed Cdc25 with an NLS to force Cdc25 into the nucleus? This would be valuable to test the model presented.

14) It would be valuable to carry out Y2H to define the physical interaction of Kpp2 with Cdc25 and also between Pcl12, CRK1, Kap123 and Cdc25. Have these experiments been carried out?

15) Subsection “Pheromone cascade-induced cell cycle arrest is mediated by the phosphorylation of Cdc25 importin, Kap123” is the first time a definitive conclusion is made in the manuscript. This is problematic for me. The paper needs to be more simply presented and broken into conclusions that build the story logically. I am not convinced by the way, or order in which the paper is currently presented. The nuclear localisation status of cdc25 needs to be introduced as a concept much earlier, following what is already known about the b-factor cell cycle arrest mechanism. This would make much more sense than how the paper is currently written.

16) Subsection “The absence of G2 cell cycle arrest during pheromone response has little impact on the ability of *U. maydis* to infect plants”. This section doesn't logically follow the one that precedes it.

17) In Figure 1—figure supplement 2A, why do the authors not include a loading control to show that amounts of protein loaded are the same for every lane? Differences in loading of samples are corrected by dividing each phosphopeptide-specific antibody signal by the Cdk1 (anti-PSTAIRE) antibody signal. However, this only gives a relative abundance of P/Cdk1. I would prefer to see loading controls too.

In summary, I think the paper has definite merit and potential significance in explaining the link between the two distinct cell cycle arrests and the onset of virulence by U. maydis. However, it is illogically presented in my view and often the detail obscures the truly significant findings. For me, a key finding is that the pheromone-induced cell cycle arrest is on its own dispensable for virulence. However, it enables the b-dependent cell cycle arrest, which is absolutely essential. Secondly, although mechanistically the b-dependent G2 arrest and pheromone arrest differ, they actually have a conserved underlying mechanism which requires retaining Cdc25 in the cytoplasm. This needs much greater emphasis too. Finally, the paper is also very narrowly framed. Only studies in Ustilago maydis are described (or if compared, only to Sch. pombe and *S. cerevisiae* models). What about cell cycle-dependent infection by other pathogenic fungi? What happens in other smuts? Are they also G2 arrested for infection? What about Colletotrichum and Magnaporthe, where cell cycle controls have been implicated in appressorium development? Are there parallels or not? Are the mechanisms distinct in basidiomycete pathogens? Some wider context is important too. Otherwise the paper's potential main conclusions are lost in impenetrable detail.

[Editors' note: further revisions were requested prior to acceptance, as described below.]

Thank you for resubmitting your work entitled "Cytoplasmic retention and degradation of a mitotic inducer enable plant infection by a pathogenic fungus" for further consideration at *eLife*. Your revised article has been favorably evaluated by Christian Hardtke (Senior Editor), a Reviewing Editor, and two reviewers.

Before we can accept the manuscript, please detail how the nuclear GFP-Cdc25 was quantified, as requested by reviewer 1.

*Reviewer #1:*

This revised version of the manuscript is much improved: the authors have performed some of the suggested experiments and addressed most of my experimental concerns, or provided convincing arguments. The only small experimental point I would like to raise concerns the quantification of nuclear GFP-Cdc25. The images provided a more convincing, and there is some quantification given, though the quantification is not described in the methods and appears rather subjective: the authors appear to visually classify cells to have nuclear GFP-Cdc25 or not. What's the cut-off used to decide for or against nuclear localization? An objective quantification requires to measure the fluorescence intensity in nuclear and cytosolic compartments in individual cells, for instance derive a ratio, and then present the mean and deviation on a graph.

The text has also been much improved: the logical flow is better apparent, and the text is overall easier to read. However, the Abstract is still difficult to understand and suffers from poor English language.

*Reviewer #2:*

I think the authors have done a very commendable revision of the paper, addressing both my concerns and those of the other expert reviewer. I think the paper is much clearer and accessible now, which is important, and I think the response to questions about additional potential experiments are thoughtful and valid. These have been carried out or described where feasible, and credible explanations provided where they are not. I am satisfied with this. Importantly, the authors are careful to explain the limitations of their conclusion too. I am happy that this version is published. It is a valuable contribution.

---

## [Author Response]

Both reviewers agree that your work is of interest. However, they are also of the opinion that your data do not yet unequivocally support your interpretation, especially the novelty aspect of temporarily separate G2 blocks that are achieved by diverse mechanisms. It is essential that you address these issues comprehensively in a revised version of your manuscript.We also all agreed that your manuscript requires extensive re-writing to make your study more easily accessible and comprehensible for the average reader. In its current form, we all found the story difficult to follow.Finally, please do your best to address, wherever possible within a reasonable time frame, the other comments in the reviews pasted below.Reviewer #1:In this manuscript, the authors provide a large amount of data addressing the mechanisms by which the corn smut Ustilago maydis arrests the cell cycle in G2 in response to pheromone during and after mating, which precedes plant colonization. Their main findings are (1) that activation of the pheromone-MAPK cascade leads to retention of the Cdc25 phosphatase in the cytosol due to phosphorylation of the Cdc25 importin likely by a complex formed between the cyclin Pcl12 and the CDK-like kinase Crk1, and (2) high-level activation of the pheromone-MAPK kinase leads to downregulation of Cdc25 protein levels due to phosphorylation of the N-terminal tail of Cdc25 likely by the MAPK Kpp2. Interestingly, preventing the cell cycle block by blocking importin phosphorylation does not prevent mating, nor plant pathogenesis. By contrast, preventing Cdc25 downregulation has no effect on pre-fusion cell cycle arrest, but impairs post-fusion cell cycle block and pathogenesis. The data is interpreted in terms of importance of this dual regulation during a cell fate change.This is an interesting story, for which a lot of the data is solid, with genetics and biochemistry supporting the claims. There are, however, a few areas that need to be looked at to make it a strong paper for publication.Experiments/data analysis:The figures showing the localization of GFP-Cdc25 are not convincing (except Figure S10C). Providing both single channels and merge would help, maybe of single examples at higher magnification. This should be coupled with quantification of nuclear and cytosolic levels.

We have included single channels in each figure (Figure 3A, Figure 4F, Figure 5C) as well as quantification of filaments showing nuclear GFP signal. Also, as reviewer suggested, we included, in some of the images, higher magnification of selected merged nuclei (Figure 3A). We thank the reviewer for the suggestion.

One important message from the paper is that Cdc25 is regulated by two distinct mechanisms: a cytosolic retention mechanism keeping pre-mating cells in G2, and a degradation mechanism important to keep post-mating cell in G2. From the data using mating cells, I am not fully convinced that this distinction in temporality can be made. First, the authors claim that mating appears to proceed normally in the kap123-T867A mutant, which fails to block the cell cycle in G2 when fuz7-DD or pcl12 are overexpressed. However, Figure 4A, showing Kap123 migration pattern in response to pheromone stimulation, is not convincingly showing the claimed mobility shift. The gel appears to be slightly smiley, but I don't see bands migrating slower. This lowers the confidence that Kap123 is in fact phosphorylated in these conditions.

We agree the reviewer that the shift on electrophoretic mobility is not impressive.

The electrophoretic shift is not present in regular SDS gels (we use BioRad TGX 4-20%). When we initiated this study, we tried PhosTag gels, but after a considerable effort with no success, we tried empirical manipulation of the ratio of acrylamide to bisacrylamide, which has been reported by different groups to improve resolution of differently phosphorylated forms of a protein. The best condition we found was 8% acrylamide/0.1% bisacrylamide. Regarding the claimed gel in old Figure 4A, we repeated this gel but alternating the lanes for control and mutant samples, conditions that help to detect more clearly the mobility shift. This new gel is now in Figure 5B. We hope this new gel helps to convince the reviewer about this mobility shift.

In addition, though kap123-T867A mutants form filaments that often have >2 nuclei in mating conditions (Figure 4D), it is not clear whether this results from a pre-mating failure in G2 block as claimed by the authors, or a post-mating block failure. In fact, when assessing the cdc25-AAA mutant, the authors perform exactly the same experiment (Figure 6C) and interpret the multi-nucleated defect as a post-fusion cycling issue. Imaging the filaments pre- and post-fusion in these conditions would address the issue.

The reviewer is right in the sense that in both experiments the result seems to be the same (multinucleated filaments), but we consider the defect that cause this phenotype to be different in each case. We base our conclusion in two main differences between these two results. The first one is that while the cdc25AAA mutant does not affected the cell cycle arrest induced by the activation of pheromone MAPK cascade (Figure 7H), the kap123-T867A mutant disables the pheromone-dependent cell cycle arrest, so we believe that in this last case the observed defects most-likely are pre-fusion. The second difference is the total number of nuclei that we found in the respective mutant filaments. While in kap123-T867A mutant crosses we never found filaments with more than 4 nuclei each (which can be explained by the establishment of a G2 barrier by the b factor once the fusion takes place, -in the text we described in a detailed way how this barrier could explain these results-), in the case of cdc25AAA crosses, we found filaments with more than 4 nuclei, suggesting that the b-dependent cell cycle arrest was not functional. We detailed the number of nuclei per filaments in both cases to made more evident these differences (new Figures6D and Figure 8D). In addition, we tried to clarify these differences along the text in the revised version.

Similarly, because Cdc25 levels decrease with increasing dosage of pheromone, the authors hypothesize that Cdc25 is a low-affinity substrate for the MAPK (subsection “The MAPK Kpp2 directs Cdc25 downregulation”), which is phosphorylated and destabilized only upon high pheromone signal. However, the authors do not test whether the cytosolic retention of Cdc25 is also pheromone-level dependent.

We found that the lowest pheromone concentration was still able to arrest the cell cycle (Figure 5A), to induce retarded electrophoretic mobility of Kap123 (Figure 5B) and to avoid the nuclear localization of Cdc25 (Figure 5D). However, since the same amount of pheromone did not result into decrease of Cdc25 levels (Figure 7A) we believe that retention of Cdc25 (and thereby cell cycle arrest) occurs even a low pheromone concentration but that the proposed degradation of Cdc25 needs higher signaling through the MAPK cascade. The reviewer is right in the sense that these results does not directly implies Cdc25 to be a low-affinity substrate and therefore we removed this sentence. At least, these results suggest that some threshold of MAPK signaling is required to induce the decrease of Cdc25 levels.

It could equally be that Kap123 importin phosphorylation scales with pheromone concentration and that Cdc25 is retained more in the cytosol in high pheromone levels, which will lead to more Cdc25 phosphorylation and degradation given this only happens in this compartment. These issues need to be clarified.

The reviewer is right and it seems that the proposed degradation of Cdc25 needs the protein to be retained at the cytoplasm (Figure 7B), but this is not the only requisite, since it also needs signaling through the MAPK cascade (Figure 7C). In the revised version, we explained this in a clearer way.

It is unfortunate that the paper just falls short of demonstrating a direct substrate-kinase relationship between the importin and Crk1 and between Cdc25 and Kpp2, though it shows existence of complexes between the importin and Ltc12, and between Cdc25 and Kpp2.

We believe that physical interactions are supported by the genetics results, but reviewer is correct, and we are aware of this weakness. Therefore, we are cautious with our conclusions about the complex. We tried along the time invested in this study other approaches to show direct substrate-kinase relationships, such as 2-yeast hybrid or in vitro assays, but they were not successful, mainly because technical problems (see answers to reviewer 2).

Text changes:The writing needs to be looked at. From the abstract it is not clear whether there are two distinct cell cycle arrests during distinct cycles, or two distinct mechanisms of arrest during the same cell cycle. It is only after reading the paper that I understood that the two arrests refer to two distinct mechanisms that occur during mating of haploid cells pre-fusion and in dikaryotic cells post-fusion, which together maintain the cell in the same G2 phase. The same goes in other parts of the manuscript. For instance, "the mutant filaments" (subsection “The pheromone-dependent decrease in the Cdc25 level enables b dependent cell cycle arrest.”) could be more clearly spelled out as dikaryotic post-fusion cells. There are also a number of grammatical mistakes.

The reviewer is correct, and we changed substantially the text to be more clear.

We also asked a professional editing system to edit the text for English.

In the Discussion section, I do not understand in what way repression of clb1 transcription by Biz1 would be irreversible. The discussion spends much time discussing these regulatory mechanisms, which are not the object of this paper, but does not discuss some pertinent questions raised by the data, for instance why Cdc25 levels do not drop upon Pcl12 over-expression but Pcl12 is required for the Cdc25 levels drop upon fuz7-DD expression. My guess is that Cdc25 downregulation can only happen in the cytosol and that Pcl12 is required for Cdc25 cytosolic retention, but this is not explicitly formulated.

The reviewer is again correct and in this new version we followed her/his advice.

Thanks for it.

Reviewer #2:Cytoplasmic Retention and degradation of a mitotic inducer enable the steps for plant infection by a pathogenic fungus. Bardetti et al.This study seeks to determine the mechanism of cell cycle control that leads to plant infection by the corn smut fungus Ustilago maydis. During infection, the fungus forms a dikaryon, which is the infective state. This requires synchrony between the cell cycles of the fusing haploid sporidia, so that subsequent development is co-ordinated. There are two discrete cell cycle arrests. The first is initiated upon pheromone recognition by each mating partner, while a second cell cycle arrest occurs after cell fusion leading to establishment of the infective dikaryotic filament. The significance of this study is that the authors propose that the cell cycle arrests occur by two distinct mechanisms. Furthermore, the first pheromone-induced cell cycle arrest results from inhibition of the nuclear import of the mitotic inducer Cdc25, based on inhibiting the function of the importin, Kap123. This is a novel mechanism, distinct from what is known in the model yeasts. Interestingly, the absence of the first G2 cell cycle arrest is dispensable for virulence, while the second cell cycle control point is essential. Linking the two control mechanisms is therefore necessary.The paper is interesting because it reveals a new mechanistic insight into the control of virulence by a dikaryotic basidiomycete pathogen. This builds upon the pioneering work of the Kahmann lab, in particular (plus Kronstad, Banuett and others) in working out how pheromone signalling and self/non-self recognition via bE/bW mating type locus functions. The link to cell cycle control is, however, vital, but often overlooked, so this paper is potentially significant, building on some excellent previous work by lead author Perez-Martin. Furthermore, the work is carefully executed in the main. What really lets it down (but which I hope can be remedied) is the way the paper is written. The manuscript is quite impenetrable in parts and needs to be re-drafted to include a set of intermediate conclusions at each stage in which the work is presented. This is absolutely vital, because currently I think the number of people who will understand this work, let alone appreciate its significance, is currently very small.The Abstract and Introduction section describe the conclusions in a relatively straightforward way, including the key finding that the pheromone-induced G2 arrest is a consequence of inhibition of nuclear transport of Cdc25 via Kap123. I think the key point that is missed in the Results section is the comparison with established mechanism by which b-factor induces G2 cell cycle arrest. I think the opening of the Results section should be re-written in a way that structures more tightly to the three effects by which b-factor mediates the sustained cell cycle arrest? This will help to explain the rationale for the study and the way in which the experiments were carried out. It will, for example, help to point out the differences in relation to the results of cdc25 transcriptional levels found in Figure S3B with protein levels reported in Figure 1E and Figure 1F, that are important to understand if the rest of the paper is to be understood.

The reviewer is right, and we followed her/his advices and tried to make the manuscript more readable. We thank the reviewer for thoughtful comments on the manuscript and supportive remarks.

Essential revisions:

*1) In each section, the authors need to make a firm conclusion to help orient the reader. This is almost entirely missing, which makes the manuscript really difficult to follow. In the subsection” Activation of the pheromone cascade promotes G2 cell cycle arrest via inhibitory phosphorylation of Cdk1”, the rationale for using the fuz7DD mutant, for* instance, is not fully explained and there is no conclusion provided to subsection “Activation of the pheromone cascade promotes G2 cell cycle arrest via inhibitory phosphorylation of Cdk1” at all.

We added a phrase with the conclusion at the end of each paragraph. Also, we tried to explain more carefully the rationale to use the fuz7^DD^ allele (subsection “Activation of the pheromone cascade promotes G2 cell cycle arrest via inhibitory phosphorylation of Cdk1”).

2) The introduction of the subsection “Pheromone cascade-induced cell cycle arrest depends on an alternative cyclin interacting with a Cdk-like kinase” has no introduction and does not follow logically from the previous section. The rationale for this part of the paper is not introduced until later in this subsection. This needs to come sooner, and with a better explanation.

We tried to explain the reason for study the Pcl12 protein in this context.

3) In Figure 1, I would prefer to see discrete data points rather than the bar charts shown in 1C and D.

We added the source data for all the graphs in Figures.

4) The Cdc25 overexpression experiment (subsection “Activation of the pheromone cascade promotes G2 cell cycle arrest via inhibitory phosphorylation of Cdk1”) needs more emphasis, as it is critical for understanding why the nuclear localisation idea was followed. I think some of this data should be in Figure 1 and not in Figure 1—figure supplement 4.

We agree the reviewer that this result needs more emphasis. In the new revised version, we tried our best (subsection “The molecular mechanisms for pheromone cascade-mediated cell cycle arrest are likely to be different from those described for b-dependent cell cycle arrest”).

5) Figure 2A- please add a micrograph of pcl12.

We added a micrograph of pcl12D mutant.

6. Figure 2B showing Cdk1 phosphorylation in FuzDD Δpcl12. The blot in 2B seems to show less total Cdk1 and Cdk5. Instead of having Figure 3—figure supplement 1, it might be better to more clearly to show CDK1 phosphorylation in Figure 2B. Could maybe authors clarify why they show Figure 3—figure supplement 1 vs. a blot of CDK1P?

Honestly, we do not understand this question. Old figure 2B does not show Cdk1 phosphorylation but Cdc25 levels.

Figure 3—figure supplement 1 showed that interfering with the ability to produce Cdk1 inhibitory phosphorylation (by expressing a Cdk1 allele refractory to this phosphorylation) disables the cell cycle arrest which results from ectopic expression of pcl12. This result indicated that both pcl12 and fuz7^DD^ expression probably use the same mechanism to arrest cell cycle, which is relevant because in one case we observed a decrease in Cdc25 levels, while in the other not.

7. Please clarify sentences subsection “Pheromone cascade-induced cell cycle arrest depends on an alternative cyclin interacting with a Cdk-like kinase” on the role of CDK and link with MAPK and Prf1 signaling. It is not clear what authors are trying to point out here and it is not clear that this work is already published.

We changed all this part of text and now we believe it is more clear (subsection “Pheromone cascade-induced cell cycle arrest depends on an alternative cyclin interacting with a Cdk-like kinase.”)

*8) In subsection “Pheromone cascade-induced cell cycle arrest depends on an alternative cyclin interacting with a Cdk-like kinase”, the authors suggest that Pcl12 and Crk1 interact to to cause the cell cycle arrest. Can a physical interaction be demonstrated either by co-localization of these proteins in arrested vs. non-arrested cells or* in vitro *by Y2H to show if this is direct or indirect interaction. This statement really needs much better support as no evidence is provided.*

We agree the reviewer that a much better support will be desirable. During the course of this work we tried to express these proteins in bacteria to analyze the ability of these proteins to make a complex as well as to study the kinase activity. Unfortunately, in spite of a considerable effort we were not successful obtaining soluble protein in both cases.

We also tried Y2H. Actually, before to try the Mass Spec approach, we tried a Y2H search expressing Pcl12 as a bait and a 2H library from U.maydis (a gift from Prof. W. K. Holloman, Cornell University at New York) as prey. The problem was that expression of pcl12 in *S. cerevisiae* was toxic and cells were not able to grow.

9) In Figure 2E, a GFP-cdc25 control is needed to show difference between arrested vs. non-arrested cells.

We do not included images of non-induced cells, but an image of GFP-Cdc25 in control cells growing in non-inducing conditions can be observed in Figure 3—figure supplement 2C.

10) Figure 2 we understand that Cdc25 degradation depends on Pcl12 and independent on crk1. But this conclusion reads diluted within the text. For example, the conclusion comes in the end of subsection “Pheromone cascade-induced cell cycle arrest is mediated by the phosphorylation of Cdc25 importin, Kap123” all together but maybe authors could slowly build up during the progression of the text this conclusion.

Decrease of Cdc25 levels is dependent on Pcl12, Crk1 and an active MAPK. We agree the author that in the former version probably this conclusion reads diluted.

We make more clear statement on this in the revised version.

11) Subsection “Pheromone cascade-induced cell cycle arrest depends on an alternative cyclin interacting with a Cdk-like kinase”. Again, there is no conclusion made to this section at all, which means that the next (really important) part of the paper on the Kap123 protein is introduced without any real reasoning.

The reviewer is right, and we changed substantially the text.

12) What happen to nuclear localization of Cdc25AAA? Has this been checked?

The retention of Cdc25 in response to pcl12 expression is not affected in the cdc25AAA allele. We added these data in Figure 7I.

13) Have the authors expressed Cdc25 with an NLS to force Cdc25 into the nucleus? This would be valuable to test the model presented.

It is a great idea. Unfortunately, from previous studies done in the laboratory many years ago (when we were studying the role of 14-3-3 in the regulation of Cdc25, see Mielnichuk and Pérez-Martin, 2008) we know that adding a canonical NLS to Cdc25 is lethal in U. maydis. We believe that the reason for that is the loss of the subcellular regulation, which seems to be really important to control premature entry into mitosis in this fungus.

14) It would be valuable to carry out Y2H to define the physical interaction of Kpp2 with Cdc25 and also between Pcl12, CRK1, Kap123 and Cdc25. Have these experiments been carried out?

For some reason, several of the proteins we were working with were toxicor produced undesirable effects when expressed in *S. cerevisiae*. As explained above, expression of pcl12 was toxic in S. cerevisae. crk1 (which is the homologue to Ime2) can be expressed only using the kinase dead version and kap123 expression resulted in cell cycle arrest in *S. cerevisiae* (probably because interferes with appropriated nuclear transport).

15) Subsection “Pheromone cascade-induced cell cycle arrest is mediated by the phosphorylation of Cdc25 importin, Kap123” is the first time a definitive conclusion is made in the manuscript. This is problematic for me. The paper needs to be more simply presented and broken into conclusions that build the story logically. I am not convinced by the way, or order in which the paper is currently presented. The nuclear localisation status of cdc25 needs to be introduced as a concept much earlier, following what is already known about the b-factor cell cycle arrest mechanism. This would make much more sense than how the paper is currently written.

We followed the reviewer advice and tried to present the results in a clearer way.

16) Subsection “The absence of G2 cell cycle arrest during pheromone response has little impact on the ability of U. maydis to infect plants”. This section doesn't logically follow the one that precedes it.

We tried to explain the reasoning for these experiments.

17) In Figure 1—figure supplement 2A, why do the authors not include a loading control to show that amounts of protein loaded are the same for every lane? Differences in loading of samples are corrected by dividing each phosphopeptide-specific antibody signal by the Cdk1 (anti-PSTAIRE) antibody signal. However, this only gives a relative abundance of P/Cdk1. I would prefer to see loading controls too.

In this experiment, what we are actually trying to determine is the level of inhibitory phosphorylation in each lane (corresponding to each specific condition and time). We are not comparing the signal from one to the other lane. For that, we have to divide each phosphopeptide-specific antibody signal by the corresponding Cdk1 (anti-PSTAIRE) antibody signal for each lane. Immunoblots were incubated successively first with anti-Cdc2-Y15P and after quantification and stripping with anti-PSTAIRE.

In summary, I think the paper has definite merit and potential significance in explaining the link between the two distinct cell cycle arrests and the onset of virulence by U. maydis. However, it is illogically presented in my view and often the detail obscures the truly significant findings. For me, a key finding is that the pheromone-induced cell cycle arrest is on its own dispensable for virulence. However, it enables the b-dependent cell cycle arrest, which is absolutely essential. Secondly, although mechanistically the b-dependent G2 arrest and pheromone arrest differ, they actually have a conserved underlying mechanism which requires retaining Cdc25 in the cytoplasm. This needs much greater emphasis too. Finally, the paper is also very narrowly framed. Only studies in Ustilago maydis are described (or if compared, only to Sch. pombe and S. cerevisiae models). What about cell cycle-dependent infection by other pathogenic fungi? What happens in other smuts? Are they also G2 arrested for infection? What about Colletotrichum and Magnaporthe, where cell cycle controls have been implicated in appressorium development? Are there parallels or not? Are the mechanisms distinct in basidiomycete pathogens? Some wider context is important too. Otherwise the paper's potential main conclusions are lost in impenetrable detail.

In the revised version we tried to broad our discussion to other pathogenic fungi as the reviewer suggested. Thanks so much for all suggestions and comments.

[Editors' note: further revisions were requested prior to acceptance, as described below.]

Before we can accept the manuscript, please detail how the nuclear GFP-Cdc25 was quantified, as requested by reviewer 1.Reviewer #1:This revised version of the manuscript is much improved: the authors have performed some of the suggested experiments and addressed most of my experimental concerns, or provided convincing arguments. The only small experimental point I would like to raise concerns the quantification of nuclear GFP-Cdc25. The images provided a more convincing, and there is some quantification given, though the quantification is not described in the methods and appears rather subjective: the authors appear to visually classify cells to have nuclear GFP-Cdc25 or not. What's the cut-off used to decide for or against nuclear localization? An objective quantification requires to measure the fluorescence intensity in nuclear and cytosolic compartments in individual cells, for instance derive a ratio, and then present the mean and deviation on a graph.

The reviewer is right in the sense that quantification should derive a ratio between the fluorescent signal at nuclear and cytosolic compartments and then present that numbers in a graph. In fact, we did that approach using Image J. However, we found that in some filaments the signal in cytoplasm was as strong as the signal at nucleus, and then just looking at the ratio between both signals, they were sorted as no nuclear GFP, when actually they were positive for GFP at nucleus. For that reason, we decided to plot fluorescent profile for each channel and in each nucleus and to decide as positive or not for GFP at the nucleus just looking that profile. We are aware that it could be more subjective that numbers, but the true is that this way it was more accurate that just looking at the ratio of fluorescence. We are including Author response image 1, Author response image 2, Author response image 3 and Author response image 4 indicating with asterisks those cases in which the ratio indicated no presence in the nucleus but they have clear signal of GFP inside the nucleus, just to illustrate our reasoning (The black bar inside the graphs marks the nucleus).

We have also included in Materials and methods section a description of the method to determine positive and negative cells for Cdc25 at nucleus, as reviewer was also correct in that it was not well described in the previous version. Thanks for the comment.

The text has also been much improved: the logical flow is better apparent, and the text is overall easier to read. However, the Abstract is still difficult to understand and suffers from poor English language.

We sent the previous version to a professional editor since English is not our native language. We tried our best with the Abstract.

**Author response image 2. respfig2:** 

**Author response image 3. respfig3:** 

**Author response image 4. respfig4:**